# Sparse Mean Estimation in Adversarial Settings via Incremental Learning

**Jianhao Ma**  *jianhao@umich.edu*
*University of Pennsylvania*

**Rui Ray Chen**  *chenrui20@mails.tsinghua.edu.cn*
*Tsinghua University*

**Yinghui He**  *yh0068@princeton.edu*
*Princeton University*

**Salar Fattahi**  *fattahi@umich.edu*
*University of Michigan*

**Wei Hu**  *vvh@umich.edu*
*University of Michigan*

**Reviewed on OpenReview:** *https://openreview.net/forum?id=S3e7ikEZfg*

## Abstract

In this paper, we study the problem of sparse mean estimation under adversarial corruptions, where the goal is to estimate the $k$-sparse mean of a heavy-tailed distribution from samples contaminated by adversarial noise. Existing methods face two key limitations: they require prior knowledge of the sparsity level $k$ and scale poorly in high-dimensional settings. We propose a simple and scalable estimator that addresses both challenges. Specifically, it learns the $k$-sparse mean without knowing $k$ in advance and operates in near-linear time and memory with respect to the ambient dimension. Under a moderate signal-to-noise ratio, our method achieves the optimal statistical rate, matching the information-theoretic lower bound. Extensive simulations corroborate our theoretical guarantees. At the heart of our approach is an *incremental learning* phenomenon: we show that a basic subgradient method applied to a nonconvex two-layer formulation with an $\ell_1$-loss can incrementally learn the $k$ nonzero components of the true mean while suppressing the rest. More broadly, our work is the first to reveal the incremental learning phenomenon of the subgradient method in the presence of heavy-tailed distributions and adversarial corruption.

## 1 Introduction

Almost all statistical methods rely explicitly or implicitly on certain assumptions on the distribution of the data. In practice, however, these assumptions are only approximately satisfied, mainly due to the presence of heavy-tailed distributions and adversarial corruptions (Rousseeuw et al., 2011). To resolve these issues, the field of robust statistics has been developed to construct estimators that exhibit "*insensitivity to small deviations from the (model) assumptions*" (Huber, 2011, p.2). Robust statistics has a long history with the fundamental work of John Tukey (Tukey, 1960; 1962), Peter Huber (Huber, 1964; 1967), and Frank Hampel (Hampel, 1971; 1974). It has been applied across various domains, such as biology, finance, and computer science (Rousseeuw et al., 2011).

Nonetheless, in high-dimensional scenarios, robust statistics contends with the *curse of dimensionality*. Firstly, the majority of estimators in the literature demand exponential runtime with respect to data dimension. To resolve this problem, special attention has been devoted to *algorithmic robust statistics*, which

aims to design efficient algorithms for different tasks in the high-dimensional robust statistics (see the recent book (Diakonikolas & Kane, 2023) and survey paper (Diakonikolas & Kane, 2019)). Secondly, generic high-dimensional robust statistical tasks are often oblivious to the intrinsic structure of the data. As such, they rely on overly conservative sample sizes that have an undesirable dependency on the data dimension.

In this paper, we aim to address these challenges for one of the most fundamental problems in robust statistics, namely *robust sparse mean estimation*. More specifically, given an $\epsilon$-corrupted set of samples from an unknown and possibly heavy-tailed distribution $\mathbb{P}$ with a $k$-sparse mean $\boldsymbol{\mu}^\star = \mathbb{E}[X] \in \mathbb{R}^d$, our goal is to design a computationally and statistically efficient estimator $\hat{\boldsymbol{\mu}}$ of the mean $\boldsymbol{\mu}^\star$. Throughout this paper, we focus on the so-called *strong contamination model* (Diakonikolas & Kane, 2023, Definition 1.6) for the corruption in the data, which encompasses a variety of existing models, such as Huber's contamination model (Huber, 1964).

**Definition 1** (Strong contamination model)**.** *Given a corruption parameter $\epsilon \in (0, \epsilon_0)$ and distribution $\mathbb{P}$, the $\epsilon$-corrupted samples are generated as follows: (i) the algorithm specifies the number of samples $n$ and then $n$ i.i.d. samples are drawn from $\mathbb{P}$. (ii) An arbitrarily powerful adversary then inspects the samples, removes $\epsilon n$ of them, and replaces them with arbitrary points. The resulting $\epsilon$-corrupted samples are given to the algorithm.*

Designing a statistically and computationally efficient estimator for the mean is highly nontrivial in this setting due to the following reasons. First, contrary to the robust (dense) mean estimation, there is a conjectured *computational-statistical tradeoff* (Diakonikolas et al., 2017b; Brennan & Bresler, 2019; 2020) for the robust $k$-sparse mean estimation, which asserts that any efficient algorithm needs $\tilde{\Omega}(k^2)$ samples, while its statistically-optimal (but possibly inefficient) counterpart only requires $\tilde{\mathcal{O}}(k)$ samples. This conjecture has neither been proved nor refuted. Second, most existing mean estimators are designed for light-tailed distributions (Balakrishnan et al., 2017; Diakonikolas et al., 2019b; Cheng et al., 2021). The only two efficient estimators available for heavy-tailed distributions (Diakonikolas et al., 2022b;a), however, are impractical for real-world applications, as they rely on computationally intensive techniques such as the ellipsoid algorithm and the sum-of-squares method. A fundamental question thus arises:

> *Can we design a practically efficient estimator for the robust sparse mean estimation problem that overcomes the conjectured computational-statistical tradeoff?*

In this work, we provide an affirmative answer to this question under moderate assumptions. Our proposed approach comprises two stages. In the first stage, we provide a coarse-grained estimation of the mean that is enough to identify the top-$k$ nonzero elements of the mean. In particular, we show that a simple subgradient method applied to a two-layer diagonal linear neural network with $\ell_1$-loss can identify the top-$k$ nonzero elements of the mean incrementally and sequentially while keeping the zero entries arbitrarily small. After the identification of the top-$k$ nonzero elements, in the second stage, we provide a finer-grained estimation of the nonzero elements of the mean by employing a generic robust mean estimator—such as those introduced in Diakonikolas & Kane (2019); Cheng et al. (2020)—restricted to the top-$k$ nonzero elements, thereby reducing the effective dimension of the problem from $d$ to $k$. Our proposed approach achieves optimal statistical error, sample complexity, and computational cost under moderate assumptions. Furthermore, we demonstrate that these assumptions do not alter the inherent complexity of the problem, as evidenced by a matching information-theoretic lower bound. Table 1 provides a summary of our results compared to the existing estimators. Our contributions are summarized below:

- **Overcoming the *computational-statistical tradeoff*.** We demonstrate that our algorithm can surpass the conjectured *computational-statistical tradeoff* under additional conditions. At a high level, we require an $\epsilon$-dependent upper bound for the coordinate-wise third moment and a lower bound for the signal-to-noise ratio (SNR). Additionally, we demonstrate that our algorithm matches the information-theoretic lower bound under exactly the same conditions.

- **Near-linear dependency on the dimension.** The first stage of our algorithm is coordinate-wise decomposable and fully parallelizable. Therefore, it runs in $\tilde{\mathcal{O}}(d)$ time and memory on a single thread, and in $\tilde{\mathcal{O}}(d/K)$ time and $\tilde{\mathcal{O}}(d)$ memory on $K$ threads. Moreover, the computational cost of

| Algorithm | $\ell_2$-error | Sample complexity | Running time |
|---|---|---|---|
| Lower bound | $\Omega(\sqrt{\epsilon})$ | $\tilde{\Omega}(k/\epsilon)$ | - |
| (Depersin, 2020; Prasad et al., 2020a) | $\mathcal{O}(\sqrt{\epsilon})$ | $\tilde{\mathcal{O}}(k/\epsilon)$ | $\exp(d)$ |
| (Diakonikolas et al., 2022b) | $\mathcal{O}(\sqrt{\epsilon})$ | $\tilde{\mathcal{O}}\left(k^2/\epsilon\right)$ | $\mathrm{poly}(d)$ |
| (Diakonikolas et al., 2022a) | $\mathcal{O}(\sqrt{\epsilon})$ | $\tilde{\mathcal{O}}(k^{\mathcal{O}(1)}/\epsilon)$ | $\mathrm{poly}(d)$ |
| Ours (Stage 1)* | $\mathcal{O}(\sqrt{k\epsilon})$ | $\tilde{\mathcal{O}}(1/\epsilon)$ | $\tilde{\mathcal{O}}(d)$ |
| Ours (full)* | $\mathcal{O}(\sqrt{\epsilon})$ | $\tilde{\mathcal{O}}(k/\epsilon)$ | $\tilde{\mathcal{O}}(d)$ |

Table 1: Comparisons between different algorithms for robust sparse mean estimation. Here, $k$ represents the sparsity level, $d$ is the ambient dimension, and $\epsilon$ denotes the corruption ratio. We use $\tilde{\Omega}(\cdot)$ and $\tilde{\mathcal{O}}(\cdot)$ to hide logarithmic factors. For simplicity, the dependency on the sample size is omitted in the above comparisons. *Our algorithms require some mild assumptions as detailed in Theorem 1.

the second stage of our algorithm is independent of $d$. In contrast, the existing robust sparse mean estimators have a poor dependency on $d$ (see Table 1).

- **No prior knowledge on the sparsity level.** Our method does not require prior knowledge of the sparsity level $k$. In contrast, all existing methods for robust sparse mean estimation (in both light- and heavy-tailed settings) require knowledge of the sparsity level $k$.

- **Superior practical performance.** Through extensive experiments, we show that, despite its simplicity, our method performs well across a broad class of heavy-tailed distributions, including those with unbounded variance.

## 2 Related Work

**Robust (sparse) mean estimation.** Robust mean estimation is a fundamental problem in statistics, with its earliest work dating back to Tukey (1960); Huber (1964). However, throughout its extensive history (Yatracos, 1985; Donoho & Liu, 1988; Donoho & Gasko, 1992; Huber, 2011), and even up to recent times (Lugosi & Mendelson, 2019b;c; Depersin, 2020; Prasad et al., 2020a), most statisticians have primarily focused on developing statistically optimal estimators, often overlooking the fact that these estimators can be computationally inefficient. It is only recently, following the seminal work of Lai et al. (2016); Diakonikolas et al. (2019a), that researchers have started to develop polynomial-time algorithms for robust mean estimation (Diakonikolas et al., 2017a; Steinhardt et al., 2017; Cheng et al., 2019a) as well as other robust learning tasks, including robust PCA (Balakrishnan et al., 2017) and robust regression (Chen et al., 2013).

Robust sparse mean estimation, as a distinct variant, has attracted considerable attention, particularly in extremely high-dimensional settings. However, the situation for robust sparse mean estimation is more nuanced compared to the dense case. Firstly, unlike the dense case, there is a conjectured *computational-statistical tradeoff* (Diakonikolas et al., 2017b; Brennan & Bresler, 2019; 2020), suggesting that efficient algorithms demand a qualitatively larger sample complexity than their inefficient counterparts. In particular, there is evidence that such a tradeoff is unavoidable for Stochastic Query (SQ) algorithms (Diakonikolas et al., 2017b). On the other hand, most prior works have primarily concentrated on the light-tailed setting (Balakrishnan et al., 2017; Diakonikolas et al., 2019b; Cheng et al., 2021). Researchers have only recently addressed the heavy-tailed setting using stability-based approaches (Diakonikolas et al., 2022b) and sum-of-squares methods (Diakonikolas et al., 2022a). While these algorithms are polynomial-time, they may not be practical when dealing with high-dimensional settings.

**Incremental learning.** Over the past few years, it has been shown practically and theoretically that gradient-based methods tend to explore the solution space in an incremental order of complexity, ultimately favoring low-complexity solutions in numerous machine learning tasks (Gissin et al., 2019; Ma et al., 2025). This phenomenon is known as *incremental learning*. Specifically, researchers have investigated incremental

learning in various contexts, such as matrix factorization and its variants (Li et al., 2020; Ma et al., 2022; Jin et al., 2023), tensor factorization (Razin et al., 2021; 2022; Ma et al., 2022), deep linear networks (Arora et al., 2019; Gidel et al., 2019; Li et al., 2021; Ma & Fattahi, 2022), and general neural networks (Hu et al., 2020; Frei et al., 2022). In essence, incremental learning is believed to be crucial for understanding the empirical success of optimization and generalization in contemporary machine learning (Gissin et al., 2019). However, to the best of our knowledge, its emergence in adversarial settings remains unexplored.

**Notation:** We use the notations $a(n) \lesssim b(n)$ and $a(n) = \mathcal{O}(b(n))$ to denote $a(n) \leq Cb(n)$, for a universal constant $C$ and sufficiently large $n$. Similarly, the notations $a(n) \gtrsim b(n)$ and $a(n) = \Omega(b(n))$ are used to denote $a(n) \geq Cb(n)$, for a universal constant $C$ and sufficiently large $n$. The notation $a = \Theta(b)$ is used to denote $a = \mathcal{O}(b)$ and $b = \mathcal{O}(a)$. Moreover, the notation $a(n) = o(b(n))$ implies that $\lim_{n \to +\infty} a(n)/b(n) = 0$. The $\text{sign}(\cdot)$ function is defined as $\text{sign}(x) = x/|x|$ if $x \neq 0$, and $\text{sign}(0) = [-1, 1]$. We also define $\widetilde{\text{sign}}(x) = x/|x|$ if $x \neq 0$, and $\widetilde{\text{sign}}(0) = 0$. Given a set $\mathcal{X}$, the indicator function $\mathbb{I}_{\mathcal{X}}(\cdot)$ is defined as $\mathbb{I}_{\mathcal{X}}(x) = 1$ if $x \in \mathcal{X}$, and $\mathbb{I}_{\mathcal{X}}(x) = 0$ otherwise. Similarly, and with a slight abuse of notation, for an event $\mathcal{E}$, we define the indicator function $\mathbb{I}(\mathcal{E}) = 1$ if $\mathcal{E}$ occurs, and $\mathbb{I}(\mathcal{E}) = 0$ otherwise. We denote $[n] := \{1, 2, \cdots, n\}$. For two functions $f, g : \mathbb{R}^d \to \mathbb{R}$, we define $\|f - g\|_\infty = \sup_{x \in \mathbb{R}^d} |f(x) - g(x)|$. For two vectors $x, y \in \mathbb{R}^d$, their Hadamard product is defined as $x \odot y = [x_1 y_1 \ \cdots \ x_d y_d]^\top$. For a vector $x \in \mathbb{R}^d$, we define $x^2 = [x_1^2, \cdots, x_d^2]^\top$. For a vector $x \in \mathbb{R}^d$ and index set $I$ with size $k$, the notation $[x]_I \in \mathbb{R}^k$ refers to the projection of $x$ onto $I$. Moreover, we define $x \wedge y = \min\{x, y\}$. We represent mixtures of probability distributions as linear combinations of their corresponding density functions. For example, given two distributions $\mathbb{P}_1$ and $\mathbb{P}_2$ and a scalar $0 \leq \epsilon \leq 1$, we define the mixture $\mathbb{P}_3 = (1 - \epsilon)\mathbb{P}_1 + \epsilon\mathbb{P}_2$. A sample from $\mathbb{P}_3$ is drawn from $\mathbb{P}_1$ with probability $1 - \epsilon$ and from $\mathbb{P}_2$ with probability $\epsilon$.

## 3 Overview of Our Approach

To lay the groundwork, we begin by introducing the standard *median-of-means* (MoM) estimator (Nemirovskij & Yudin, 1983; Jerrum et al., 1986; Alon et al., 1996) originally designed for estimating the mean of a one-dimensional random variable. The MoM estimator serves as a cornerstone for more sophisticated methods as detailed in Lugosi & Mendelson (2019c); Prasad et al. (2020b); Lecué & Lerasle (2020); Diakonikolas et al. (2022b).

**Definition 2** (Median-of-means estimator for one-dimensional case). *Given a set of $\epsilon$-corrupted samples $S = \{X_1, \cdots, X_n\} \subset \mathbb{R}$, we first partition them into $J$ subgroups $S_1, \cdots, S_J$ with equal sizes, where we assume $n$ is divisible by $J$ for simplicity. We then calculate the sample mean for each subgroup, i.e., $\bar{X}_j = \frac{1}{B}\sum_{i \in S_j} X_i$ where $B = n/J$. Subsequently, the median-of-means (MoM) estimator is obtained by taking the median of the sample means $\bar{X}_1, \cdots, \bar{X}_J$, i.e., $\hat{\mu}_{MoM} = \text{median}\{\bar{X}_1, \cdots, \bar{X}_J\}$.*

Alternatively, the MoM estimator can be expressed as the minimizer of the following $\ell_1$-loss:

$$\hat{\mu}_{\mathsf{MoM}} = \arg\min_{\mu \in \mathbb{R}} \frac{1}{J} \sum_{j=1}^{J} |\bar{X}_j - \mu|. \tag{1}$$

By appropriately selecting the number of subgroups $J$, it can be shown that the MoM estimator matches the information-theoretic lower bound $\Omega(\sigma\sqrt{\epsilon})$ for heavy-tailed distributions under the strong contamination model (Definition 1).

**Proposition 1** (One-dimensional MoM estimator). *Consider a corruption parameter $\epsilon \leq \frac{1}{8}$, a failure probability $\delta$, and a set $S$ of $n$ many $\epsilon$-corrupted samples from a distribution $\mathbb{P}$ with mean $\mu^\star \in \mathbb{R}$ and variance $\mathbb{E}[(X - \mu^\star)^2] \leq \sigma^2$. Suppose that $n \gtrsim \log(1/\delta)/\epsilon$. Then, upon choosing the number of subgroups $J = \Theta(\lceil \epsilon n \rceil + \log(1/\delta))$, with probability at least $1 - \delta$ over the sample set $S$, the MoM estimator $\hat{\mu}_{MoM}$ satisfies $|\hat{\mu}_{MoM} - \mu^\star| = \mathcal{O}(\sigma\sqrt{\epsilon})$.*

A more precise statement of Proposition 1 and its proof are presented in Appendix A.

Naively applying the MoM estimator to different coordinates of a high-dimensional random variable leads to an undesirable dependency on the dimension $d$. More precisely, the coordinate-wise MoM, which corresponds to the solution to the following convex optimization

$$\hat{\boldsymbol{\mu}}_{\mathsf{MoM}} = \arg\min_{\boldsymbol{\mu}\in\mathbb{R}^d} \mathcal{L}_{\mathsf{cvx}}(\boldsymbol{\mu}) := \frac{1}{J}\sum_{j=1}^{J}\left\|\bar{X}_j - \boldsymbol{\mu}\right\|_1, \qquad (\text{CVX})$$

suffers from a suboptimal error rate of $\|\hat{\boldsymbol{\mu}}_{\mathsf{MoM}} - \boldsymbol{\mu}^\star\|_2 = \mathcal{O}(\sigma\sqrt{d\epsilon})$ (see Theorem 5 in Appendix A). This error is unavoidable for the MoM estimator since the coordinate-wise error $\mathcal{O}(\sigma\sqrt{\epsilon})$ is uniformly distributed across each coordinate. An alternative approach, the geometric MoM (Minsker, 2015), which replaces the $\|\cdot\|_1$ in CVX by $\|\cdot\|_2$, also suffers from a similar error.

**Two-layer model** To address the above issue, we model the mean $\boldsymbol{\mu}$ as a two-layer model $\boldsymbol{u}^2 - \boldsymbol{v}^2$ for $\boldsymbol{u}, \boldsymbol{v} \in \mathbb{R}^d$, and obtain $(\boldsymbol{u}, \boldsymbol{v})$ by minimizing the following nonconvex $\ell_1$-loss

$$\min_{\boldsymbol{u},\boldsymbol{v}\in\mathbb{R}^d} \mathcal{L}_{\mathsf{ncvx}}(\boldsymbol{u},\boldsymbol{v}) = \frac{1}{2J}\sum_{j=1}^{J}\left\|\bar{X}_j - \left(\boldsymbol{u}^2 - \boldsymbol{v}^2\right)\right\|_1. \qquad (\text{NCVX})$$

To solve this optimization problem, we propose a subgradient method (SubGM) with small initialization $\boldsymbol{u}(0) = \boldsymbol{v}(0) = \alpha\vec{1}$, where $\vec{1} = [1, \cdots, 1]^\top \in \mathbb{R}^d$ and $\alpha > 0$ is a sufficiently small factor. At each iteration, SubGM updates the solution as

$$\begin{aligned} \boldsymbol{u}(t+1) &= \boldsymbol{u}(t) - \eta\boldsymbol{g}(t) \quad\text{where}\quad \boldsymbol{g}(t) \in \partial_{\boldsymbol{u}}\mathcal{L}_{\mathsf{ncvx}}(\boldsymbol{u}(t), \boldsymbol{v}(t)), \\ \boldsymbol{v}(t+1) &= \boldsymbol{v}(t) - \eta\boldsymbol{h}(t) \quad\text{where}\quad \boldsymbol{h}(t) \in \partial_{\boldsymbol{v}}\mathcal{L}_{\mathsf{ncvx}}(\boldsymbol{u}(t), \boldsymbol{v}(t)). \end{aligned} \qquad (\text{SUBGM})$$

Here, $\eta > 0$ is the stepsize, and $\partial_{\boldsymbol{u}}\mathcal{L}_{\mathsf{ncvx}}(\boldsymbol{u}, \boldsymbol{v})$ and $\partial_{\boldsymbol{v}}\mathcal{L}_{\mathsf{ncvx}}(\boldsymbol{u}, \boldsymbol{v})$ indicate the (Clarke) subdifferentials of $\mathcal{L}_{\mathsf{ncvx}}$, defined as:

$$\partial_{\boldsymbol{u}}\mathcal{L}_{\mathsf{ncvx}}(\boldsymbol{u}, \boldsymbol{v}) = \frac{1}{J}\sum_{j=1}^{J}\mathrm{sign}(\boldsymbol{u}^2 - \boldsymbol{v}^2 - \bar{X}_j)\odot\boldsymbol{u}, \qquad (2)$$

$$\partial_{\boldsymbol{v}}\mathcal{L}_{\mathsf{ncvx}}(\boldsymbol{u}, \boldsymbol{v}) = -\frac{1}{J}\sum_{j=1}^{J}\mathrm{sign}(\boldsymbol{u}^2 - \boldsymbol{v}^2 - \bar{X}_j)\odot\boldsymbol{v}. \qquad (3)$$

The detailed implementation of our proposed algorithm is presented in Algorithm 1.

---

**Algorithm 1** ROBUST SPARSE MEAN ESTIMATION VIA INCREMENTAL LEARNING

---

**Input:** dataset $S$, corruption parameter $\epsilon$, failure probability $\delta$, initialization scale $\alpha$, stepsize $\eta$, and number of iterations $T \in \left[\frac{2}{\eta}\log(1/\alpha), \frac{6}{\eta}\log(1/\alpha)\right]$.

1: **Stage 1 (SubGM):**
2:     **Pre-processing:** Divide the dataset into $J$ equal subgroups $S_1, \cdots, S_J$, where $J = 100\lceil\epsilon n\rceil$. Calculate the sample means $\bar{X}_j = \frac{J}{n}\sum_{i\in S_j} X_i$.
3:     **Initialization:** $\boldsymbol{u}(0) = \boldsymbol{v}(0) = \alpha\vec{1}$.
4:     **for** $t = 1, \cdots, T$ **do**
5:         Update $\boldsymbol{u}(t), \boldsymbol{v}(t)$ via SUBGM.
6:     **end for**
7:     **Identification of top-$k$ elements:** Calculate $I = \left\{i\in[d] : |u_i^2(T) - v_i^2(T)| \geq \alpha\right\}$.
8:     **Return** $\hat{\boldsymbol{\mu}}(T) = \boldsymbol{u}^2(T) - \boldsymbol{v}^2(T)$.
9: **Stage 2** (optional):
10:     Consider the projected dataset $S_k = \{[X_i]_I : X_i \in S\}$ and apply an existing robust mean estimator (e.g. those introduced in Diakonikolas & Kane (2019); Cheng et al. (2020)) to $S_k$.

---

Our key contribution is to reveal the emergence of incremental learning: we show that SubGM with small initialization learns the nonzero components (signals) long before overfitting the zero components (residuals)

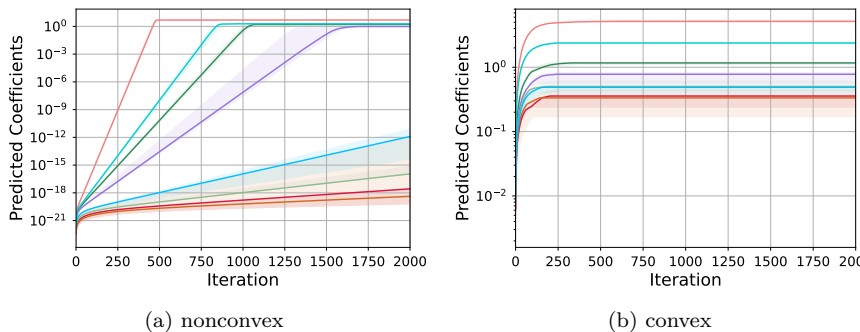

Figure 1: The predicted coefficients for NCVX and CVX. Each color corresponds to a different coordinate of the estimated mean. We run the subgradient method with stepsize $\eta = 0.07$ in both settings. We initialize NCVX with $\alpha = 1 \times 10^{-10}$ and use a zero initialization for CVX. We generate the inliers with $d = 500, n = 2000$ from a lognormal distribution with a variance of 3.3 and a $k$-sparse mean with $k = 4$ and nonzero elements $[5, 2, 2, 1.5]$. The corruption rate is 0.05 and the outliers are generated from a Cauchy distribution with location 20 and scale 50.

to noise. Consequently, there exists a wide range of iterations within which the signals are in the order of $\Omega(1)$ while the residuals remain in the order of $\mathcal{O}(\alpha)$ (see Figure 1a). Remarkably, we show that this interval only depends on the stepsize $\eta$ and the initialization scale $\alpha$, and it can be widened by reducing these user-defined parameters. In stark contrast, differentiating between the signals and residuals is challenging in the convex setting (CVX) precisely due to the lack of incremental learning, as shown in Figure 1b. After successfully identifying the locations of the top-$k$ elements, we can employ existing robust mean estimation techniques (Diakonikolas & Kane, 2019; Cheng et al., 2020) on the dataset projected onto the recovered support to further improve the estimation of the top-$k$ nonzero elements.

## 4 Main Result

In this section, we present the theoretical guarantees for Algorithm 1. We begin by analyzing the first stage of the algorithm, which focuses on recovering the support of the true mean.

### 4.1 Stage 1: Identification of Support via Coarse-grained Estimation

We denote $\mu_{\max}^\star = \max_i\{|\mu_i^\star|\}$ and $\mu_{\min}^\star = \min_i\{|\mu_i^\star| : \mu_i^\star \neq 0\}$. Our main theorem is presented next.

**Theorem 1** (Convergence guarantee for SubGM). *Let $\mathbb{P}$ be a distribution on $\mathbb{R}^d$ with an unknown $k$-sparse mean $\boldsymbol{\mu}^\star$, unknown covariance matrix $\boldsymbol{\Sigma} \preceq \sigma^2 \boldsymbol{I}$, and unknown coordinate-wise third moment satisfying $\mathbb{E}[|X_i - \mu_i^\star|^3] \lesssim \sigma^3/\sqrt{\epsilon}, \forall 1 \leq i \leq d$. Consider an arbitrary sample set of size $n \gtrsim \log(d/\delta)/\epsilon$ collected according to the strong contamination model (Definition 1) with corruption parameter $\epsilon \leq c$ where $c > 0$ is a sufficiently small universal constant. Upon setting the stepsize $\eta \leq \sigma\sqrt{\epsilon}/\mu_{\max}^\star$ and the initialization scale $0 < \alpha \lesssim \sigma\sqrt{\epsilon/d} \wedge \mu_{\max}^{\star-5}$ in Algorithm 1, with a probability of at least $1 - \delta$, the following statements hold for any iteration $\frac{2}{\eta}\log(1/\alpha) \leq T \leq \frac{6}{\eta}\log(1/\alpha)$:*

- *$\ell_2$-**error.** The $\ell_2$-error is upper-bounded by*

$$\|\hat{\boldsymbol{\mu}}(T) - \boldsymbol{\mu}^\star\|_2 \lesssim \sigma\sqrt{k\epsilon}. \tag{4}$$

- ***Identification of the top-$k$ elements.** If we additionally have $\epsilon \lesssim \mu_{\min}^{\star 2}/\sigma^2$, then we obtain*

$$\begin{aligned}
|\hat{\mu}_i(T)| &\gtrsim \sigma\sqrt{\epsilon}, &&\text{where } \mu_i^\star \neq 0, \\
|\hat{\mu}_i(T)| &\lesssim \alpha, &&\text{where } \mu_i^\star = 0.
\end{aligned} \tag{5}$$

**Comparison to the existing results.** Simply applying the coordinate-wise MoM estimator results in an $\ell_2$-error rate $\mathcal{O}(\sigma\sqrt{d\epsilon})$, which is considerably worse than our result when $k \ll d$. On the other hand, to

guarantee a correct support recovery, the previous efficient estimators rely on prior knowledge of $k$, while the coordinate-wise MoM requires an accurate value of $\mu^\star_{\min}$ to separate the signals from residuals (as evidenced by Figure 1b). In contrast, our proposed algorithm only requires a lower bound $\hat{\mu}_{\min} \leq \mu^\star_{\min}$ to differentiate the signals from residuals; in fact, this lower bound can be arbitrarily small (i.e., conservative) provided that the initialization scale is chosen as $\alpha \ll \hat{\mu}_{\min}$. We also highlight that, much like other existing estimators under the strong contamination model, our estimator requires prior knowledge of the corruption parameter $\epsilon$ (or its upper bound).

**Remark 1.** *The assumption of a coordinate-wise bounded third moment is imposed to facilitate the use of Berry-Esseen-type bounds in our analysis. This requirement is primarily technical and arises from controlling the finite-sample deviation of empirical means under contamination. We conjecture that this assumption can be relaxed or entirely removed by leveraging more refined concentration or robust mean estimation techniques. Establishing such extensions is left for future work.*

**Proof sketch.** We next provide the proof sketch of the above theorem, deferring its details to Section 5. Specifically, we analyze the coordinate-wise dynamic $\hat{\mu}_i(t) = u_i^2(t) - v_i^2(t)$ for some $1 \leq i \leq d$. Without loss of generality, we assume $\mu^\star_i \geq 0$. Upon defining $\beta_i(t) = \frac{1}{J} \sum_{j=1}^J \widetilde{\text{sign}}(\bar{X}_{j,i} - \hat{\mu}_i(t))$, the update rules for $u_i(t)$ and $v_i(t)$ can be written as

$$u_i(t+1) = (1 + \eta\beta_i(t))\, u_i(t), \quad v_i(t+1) = (1 - \eta\beta_i(t))\, v_i(t). \tag{6}$$

Based on the above update rules, $\beta_i(t)$ controls the growth rate of the dynamics. Indeed, during the initial iterations, we have $\hat{\mu}_i(t) \approx \hat{\mu}_i(0) = u_i^2(0) - v_i^2(0) = 0$, which in turn implies that $\beta_i(t) \approx \beta_i(0)$. Consequently, the dynamics of $u_i(t)$ and $v_i(t)$ can be well approximated using the following exponential functions

$$u_i(t) \approx (1 + \eta\beta_i(0))^t \alpha, \quad v_i(t) \approx (1 - \eta\beta_i(0))^t \alpha. \tag{7}$$

Therefore, to analyze the behaviors of $u_i(t)$ and $v_i(t)$, it suffices to characterize the magnitude of $\beta_i(0)$ for different coordinates. To achieve this, we define $\mathcal{J}_{\text{clean}}$ as the index set of the subgroups $[J]$ that do not contain any outliers, and denote its complement as $\mathcal{J}_{\text{outlier}} = [J]\backslash\mathcal{J}_{\text{clean}}$. We have

$$\beta_i(0) = \frac{1}{J}\sum_{j\in\mathcal{J}_{\text{clean}}}\widetilde{\text{sign}}(\bar{X}_{j,i}) \pm \frac{|\mathcal{J}_{\text{outlier}}|}{J} \qquad\qquad \text{(denote } \delta = \tfrac{|\mathcal{J}_{\text{outlier}}|}{J})$$

$$\approx (1-\delta)\mathbb{E}\left[\widetilde{\text{sign}}(\bar{X}_{j,i})\right] \pm \delta \qquad\qquad \text{(for sufficiently large } \mathcal{J}_{\text{clean}})$$

$$= (1-\delta)\left(1 - 2\Pr\left(\bar{X}_{j,i} - \mu^\star_i \leq -\mu^\star_i\right)\right) \pm \delta$$

$$\approx (1-\delta)(1 - 2\Phi(-\mu^\star_i \cdot \sqrt{B\text{Var}(X)})) \pm \delta. \qquad \text{(due to finite-sample central limit theorem)}$$

Here, $B = n/J$ is the size of each subgroup, and $\Phi(\cdot)$ represents the cumulative distribution function (CDF) of the standard Gaussian distribution. Let us define $\mathcal{I}_{\text{residual}} = \{i : \mu^\star_i = 0\}$ and $\mathcal{I}_{\text{signal}} = \{i : \mu^\star_i \neq 0\}$. Based on the above characterization of $\beta_i(0)$, for all $i \in \mathcal{I}_{\text{residual}}$, we have $1 - 2\Phi(-\mu^\star_i \cdot \sqrt{B\text{Var}(X)}) = 0$, which in turn implies $\beta_i(0) \approx \pm\delta$. Furthermore, by setting $J = C\lceil\epsilon n\rceil$ with a suitably large constant $C$, $B = n/J \geq 1/(C\epsilon)$ can be made sufficiently large given a sufficiently small $\epsilon$. This ensures that $\beta_i(0) \approx \Omega(1-\delta) \pm \delta$ for all $i \in \mathcal{I}_{\text{signal}}$. On the other hand, we have $\delta \leq \lceil\epsilon n\rceil/J \leq 1/C$ since $|\mathcal{J}_{\text{outlier}}| \leq \lceil\epsilon n\rceil$. As a result, $|\beta_i(0)|$ can be made arbitrarily small for all $i \in \mathcal{I}_{\text{residual}}$ and $\beta_i(0) = \Omega(1)$ for all $i \in \mathcal{I}_{\text{signal}}$. This discrepancy in the growth rates of $u_i(t)$ and $v_i(t)$ enables our algorithm to separate the signals from residuals within just a few iterations. In Section 5, we provide a more delicate analysis of the dynamics, showing that for all $T \in [\frac{2}{\eta}\log(1/\alpha), \frac{6}{\eta}\log(1/\alpha)]$ we have

$$\begin{aligned} u_i^2(t) - v_i^2(t) &= \mu^\star_i \pm \mathcal{O}(\sigma\sqrt{\epsilon}), &&\text{for } i \in \mathcal{I}_{\text{signal}}, \\ \left|u_i^2(t) - v_i^2(t)\right| &= \text{poly}(\alpha) \ll \sigma\sqrt{\epsilon}, &&\text{for } i \in \mathcal{I}_{\text{residual}}. \end{aligned} \tag{8}$$

The above equation sheds light on the key difference between NCVX and CVX: unlike CVX where the error is equally distributed across different coordinates, the error in NCVX is primarily distributed among the signals, while the error in the residuals can be kept arbitrarily small by a proper choice of the initialization scale $\alpha$. This implies that, if the signals are sufficiently larger than the induced error, i.e., $|\mu^\star_i| \gtrsim \sigma\sqrt{\epsilon}, \forall i \in \mathcal{I}_{\text{signal}}$, our algorithm can successfully identify the signals.

### 4.2 Stage 2: Achieving Optimal Rate on the Support via Fine-grained Estimation

As illustrated in Section 4.1, a direct application of SubGM leads to an estimation error of $\mathcal{O}(\sqrt{k}\epsilon)$. In this section, we show that this error can be further improved once the support of the mean is identified correctly. Our key insight is that once the support of the mean is recovered, we can reduce the problem to a robust *dense* mean estimation defined *only* over the recovered support. Under such a regime, existing estimators designed for robust dense mean estimation (Diakonikolas & Kane, 2019; Cheng et al., 2020) can be employed to further reduce the estimation error.

**Proposition 2** (Adapted from Proposition 1.6 in Diakonikolas et al. (2020)). *Let $\mathbb{P}$ be a distribution on $\mathbb{R}^k$ with an unknown mean $\boldsymbol{\mu}^\star$ and unknown covariance matrix $\boldsymbol{\Sigma} \preceq \sigma^2 \boldsymbol{I}$. Suppose a sample set of size $n$ is collected according to the strong contamination model (Definition 1) with corruption parameter $\epsilon < 1/2$. Then, there exists an algorithm that runs in $\mathcal{O}(kn)$ time and memory and, with a probability of at least $1 - \delta$, outputs an estimator $\hat{\boldsymbol{\mu}}$ that satisfies*

$$\|\hat{\boldsymbol{\mu}} - \boldsymbol{\mu}^\star\|_2 \lesssim \sigma\sqrt{\epsilon} + \sigma\sqrt{k/n} + \sigma\sqrt{\log(1/\delta)/n}.$$

Equipped with the above result, we next provide an end-to-end guarantee for our full algorithm.

**Theorem 2** (Guarantee for the full algorithm). *Let $\mathbb{P}$ be a distribution on $\mathbb{R}^d$ satisfying the conditions in Theorem 1. Consider an arbitrary sample set of size $n \gtrsim (k + \log(d/\delta))/\epsilon$ that is collected according to the strong contamination model (Definition 1) with corruption parameter $\epsilon \lesssim \mu_{\min}^2/\sigma^2 \wedge 1$. Then, with the choice of $\alpha = \Theta(\sigma\sqrt{\epsilon/d} \wedge \mu_{\max}^{\star-5})$ and $\eta = \Theta(\sigma\sqrt{\epsilon}/\mu_{\max}^\star)$, our full algorithm runs in $\mathcal{O}(nd\log(d))$ time and memory and, with a probability of at least $1 - \delta$, outputs an estimate $\hat{\boldsymbol{\mu}}$ that satisfies*

$$\|\hat{\boldsymbol{\mu}} - \boldsymbol{\mu}^\star\|_2 \lesssim \sigma\sqrt{\epsilon}. \tag{9}$$

Upon setting the sample size $n = \Theta\left((k + \log(d/\delta))/\epsilon\right)$, our proposed two-stage method runs in $\tilde{\mathcal{O}}(dk/\epsilon)$ time and memory and returns a solution with an error in the order of $\mathcal{O}(\sigma\sqrt{\epsilon})$. Our next theorem shows that this error is indeed information-theoretically optimal up to a constant factor and thus cannot be improved.

**Theorem 3** (Information-theoretic lower bound). *There exists a distribution $\mathbb{P}$ with $k$-sparse mean $\boldsymbol{\mu}^\star$, covariance matrix $\boldsymbol{\Sigma} \preceq \sigma^2 \boldsymbol{I}$, and coordinate-wise third moment satisfying $\mathbb{E}[|X_i - \mu_i^\star|^3] \lesssim \sigma^3/\sqrt{\epsilon}, \forall 1 \leq i \leq d$ such that, given any arbitrarily large sample set collected according to the strong contamination model (Definition 1) with corruption parameter $\epsilon$, no algorithm can estimate the mean $\boldsymbol{\mu}^\star$ with $\ell_2$-error $o(\sigma\sqrt{\epsilon})$.*

**Comparison to the existing lower bounds.** To achieve the optimal error rate, the sample complexity of our method scales linearly with the sparsity level $k$. A careful reader may realize that our sample complexity is unexpectedly smaller than the optimal sample complexity $\Omega((k\log(d/k)+\log(d/\delta))/\epsilon)$ introduced in Lugosi & Mendelson (2019a) when $k$ is sufficiently small. This is due to the additional assumptions we impose on the coordinate-wise third moment of the distribution and the corruption parameter $\epsilon$. On the other hand, it is recently shown in Diakonikolas & Kane (2019); Prasad et al. (2020a) that under the bounded third moment, the dependency of the estimation error on $\epsilon$ can be improved from $\epsilon^{1/2}$ to $\epsilon^{2/3}$. Our worse dependency on $\epsilon$ is due to our more relaxed assumption on the third moment: unlike the assumptions made in Diakonikolas & Kane (2019); Prasad et al. (2020a), our imposed upper bound on the third moment is inversely proportional to $\sqrt{\epsilon}$. Consequently, the imposed upper bound can get arbitrarily large with a smaller corruption parameter. In this extreme case where $\epsilon \to 0$, this condition can be dropped altogether.

**Key differences between the first and second stages.** We note that Stage 1 is primarily designed to provide a coarse-grained estimate and, in particular, to enable support identification via incremental learning. According to the first statement of Theorem 1, under the general moment conditions and the strong contamination model, Stage 1 guarantees an $\ell_2$-estimation error of order $O(\sigma\sqrt{k}\epsilon)$ within a suitable range of iterations. Stage 2, on the other hand, refines the estimate of the recovered support to achieve the optimal statistical rate $O(\sigma\sqrt{\epsilon})$. However, the success of Stage 2 relies on correctly recovering the support, which in turn requires a stronger signal-to-noise ratio (SNR) assumption on the contamination rate. Concretely, the support identification guarantee in the second statement of Theorem 1 requires an additional condition of the form $\epsilon \lesssim \mu_{\min}^{\star 2}/\sigma^2$, and Theorem 2 (end-to-end guarantee) inherits this requirement.

# 5 Proofs

The proofs of our main results are organized as follows. Section 5.1 presents preliminary lemmas. Section 5.2 establishes the convergence guarantee of SubGM (Theorem 1), and Section 5.3 provides the end-to-end guarantee of the full algorithm (Theorem 2). Section 5.4 derives the information-theoretic lower bound (Theorem 3), and Appendix A proves a formal variant of Proposition 1 to establish the properties of the MoM estimator.

## 5.1 Preliminaries

This section presents all the technical lemmas that will be used to prove our main results.

**Lemma 1** (Chebyshev's inequality (Vershynin, 2018, Corollary 1.2.5)). *Suppose that $X \sim \mathbb{P}$ with $\mathrm{Var}(X) < \infty$. Then, for any $\delta > 0$, we have*

$$\Pr\left(|X - \mathbb{E}[X]| \geq \delta\right) \leq \frac{\mathrm{Var}(X)}{\delta^2}. \tag{10}$$

**Lemma 2** (Hoeffding's inequality (Vershynin, 2018, Theorem 2.2.6)). *Let $X_1, \cdots, X_n$ be independent random variables such that $a_i \leq X_i \leq b_i$ almost surely. Then for all $\delta > 0$, we have*

$$\Pr\left(\sum_{i=1}^{n}(X_i - \mathbb{E}[X_i]) \geq \delta\right) \leq \exp\left\{-\frac{2\delta^2}{\sum_{i=1}^{n}(b_i - a_i)^2}\right\}. \tag{11}$$

**Lemma 3** (Dvoretzky-Kiefer-Wolfowitz Inequality (Massart, 1990)). *Let $F(t) = \Pr(X \leq t)$ be the CDF of a random variable $X$, and let $\hat{F}_n(\cdot) = \frac{1}{n}\sum_{i=1}^{n}\mathbb{I}_{(-\infty,\cdot]}(X_i)$ be the empirical CDF based on $n$ i.i.d. samples $X_1, \ldots, X_n \sim \mathbb{P}$. We have*

$$\Pr\left(\left\|\hat{F}_n - F\right\|_{\infty} \geq t\right) \leq 2e^{-2nt^2} \quad \text{for all } t \geq 0. \tag{12}$$

**Lemma 4.** *Suppose $X_1, \cdots, X_n \overset{i.i.d.}{\sim} \mathbb{P}$. Then, with probability at least $1 - \delta$ and for all $a \in \mathbb{R}$, we have*

$$\left|\frac{1}{n}\sum_{i=1}^{n}\widetilde{\mathrm{sign}}(X_i - a) - \mathbb{E}\left[\widetilde{\mathrm{sign}}(X - a)\right]\right| \leq \sqrt{\frac{2\log(2/\delta)}{n}}. \tag{13}$$

*Proof.* Note that $\widetilde{\mathrm{sign}}(X_i - a) = 1 - \mathbb{I}_{(-\infty,a]}(X_i) - \mathbb{I}_{(-\infty,a)}(X_i)$ and $\mathbb{E}\left[\widetilde{\mathrm{sign}}(X - a)\right] = 1 - \Pr(X \leq a) - \Pr(X < a)$. Therefore, we have

$$\begin{aligned}
&\left|\frac{1}{n}\sum_{i=1}^{n}\widetilde{\mathrm{sign}}(X_i - a) - \mathbb{E}\left[\widetilde{\mathrm{sign}}(X - a)\right]\right| \\
&\leq \left|\frac{1}{n}\sum_{i=1}^{n}\mathbb{I}_{(-\infty,a]}(X_i) - \Pr(X \leq a)\right| + \sup_{b < a}\left|\frac{1}{n}\sum_{i=1}^{n}\mathbb{I}_{(-\infty,b]}(X_i) - \Pr(X \leq b)\right| \\
&\leq 2\left\|\hat{F}_n - F\right\|_{\infty}.
\end{aligned} \tag{14}$$

Upon setting $t = \sqrt{\frac{1}{2n}\log\left(\frac{2}{\delta}\right)}$ in the Dvoretzky-Kiefer-Wolfowitz Inequality (Lemma 3), with probability at least $1 - \delta$ and for all $a \in \mathbb{R}$, we have

$$\left|\frac{1}{n}\sum_{i=1}^{n}\widetilde{\mathrm{sign}}(X_i - a) - \mathbb{E}\left[\widetilde{\mathrm{sign}}(X - a)\right]\right| \leq 2\left\|\hat{F}_n - F\right\|_{\infty} \leq \sqrt{\frac{2\log(2/\delta)}{n}}. \tag{15}$$

$\square$

**Lemma 5** (Berry-Esseen bound (Vershynin, 2018, Theorem 2.1.3)). *Suppose* $X_1, \cdots, X_n \overset{i.i.d.}{\sim} \mathbb{P}$, *where* $\mathbb{P}$ *has zero mean and bounded third moment, i.e.,* $\mu = \mathbb{E}[X] = 0, \rho = \mathbb{E}[|X|^3] < \infty$. *Then, upon denoting* $Z_n = \frac{\sum_{i=1}^n X_i}{\sigma\sqrt{n}}$ *where* $\sigma^2 = \mathbb{E}[X^2]$, *we have*

$$\sup_{a \in \mathbb{R}} |\Pr(Z_n < a) - \Phi(a)| \leq \frac{0.5\rho}{\sigma^3\sqrt{n}}. \tag{16}$$

*Here* $\Phi(\cdot)$ *is the CDF of standard Gaussian distribution.*

### 5.2 Proof of Theorem 1

To prove this theorem, it is essential to first establish the uniform concentration of $\frac{1}{J} \sum_{j=1}^J \widetilde{\text{sign}}(\bar{X}_{j,i} + a)$ for all $a \geq 0$.

**Lemma 6.** *Suppose* $X_1, \cdots, X_n \overset{i.i.d.}{\sim} \mathbb{P}$, *where* $\mathbb{P}$ *has zero mean, variance* $\sigma^2$, *and coordinate-wise third moment* $\rho$. *Moreover, suppose samples are generated according to the strong contamination model (Definition 1) with corruption parameter* $\epsilon$. *Suppose* $n \geq 20000 \log(2d/\delta)/\epsilon$, $J = 100\lceil\epsilon n\rceil$, *and* $\rho \leq 0.005\sigma^3/\sqrt{\epsilon}$. *Upon dividing the samples into* $J$ *equal subgroups* $S_1, \cdots, S_J$ *and denoting the empirical mean of each subgroup by* $\bar{X}_j = \frac{1}{B} \sum_{k \in S_j} X_k$, *with probability at least* $1 - \delta$, *the following statements hold*

- *For all* $a \geq 20\sigma\sqrt{\epsilon}$ *and all* $1 \leq i \leq d$, *we have:* $\frac{3}{5} \leq \frac{1}{J} \sum_{j=1}^J \widetilde{\text{sign}}(\bar{X}_{j,i} + a) \leq 1$.

- *For all* $0 \leq a \leq 0.001\sigma\sqrt{\epsilon}$ *and all* $1 \leq i \leq d$, *we have:* $-0.08 \leq \frac{1}{J} \sum_{j=1}^J \widetilde{\text{sign}}(\bar{X}_{j,i} + a) \leq 0.08$.

*Proof.* We prove the two cases separately.

*Case 1*: $a \geq 20\sigma\sqrt{\epsilon}$. We only need to prove the lower bound since the upper bound is trivial. We partition the index set of subgroups $1, \ldots, J$ into two disjoint subsets: $\mathcal{J}_{\text{clean}}$, containing all subgroups free of outliers, and $\mathcal{J}_{\text{outlier}}$, containing those with at least one outlier. Note that $|\mathcal{J}_{\text{outlier}}| \leq \lceil\epsilon n\rceil$. Therefore, we obtain

$$\frac{1}{J} \sum_{j=1}^J \widetilde{\text{sign}}(\bar{X}_{j,i} + a) \geq \frac{1}{J} \sum_{j \in \mathcal{J}_{\text{clean}}} \widetilde{\text{sign}}(\bar{X}_{j,i} + a) - \frac{\lceil\epsilon n\rceil}{J}. \tag{17}$$

Next, applying Lemma 4 and a union bound, we obtain that, with probability at least $1 - \delta$ and for all $1 \leq i \leq d$,

$$
\begin{aligned}
\frac{1}{J} \sum_{j=1}^J \widetilde{\text{sign}}(\bar{X}_{j,i} + a) &\geq \frac{|\mathcal{J}_{\text{clean}}|}{J} \left( \mathbb{E}\left[\widetilde{\text{sign}}(\bar{X}_{j,i} + a)\right] - \sqrt{\frac{2\log(2d/\delta)}{|\mathcal{J}_{\text{clean}}|}} \right) - \frac{\lceil\epsilon n\rceil}{J} \\
&= \frac{|\mathcal{J}_{\text{clean}}|}{J} \left( 1 - 2\Pr(\bar{X}_{j,i} \leq -a) - \sqrt{\frac{2\log(2d/\delta)}{|\mathcal{J}_{\text{clean}}|}} \right) - \frac{\lceil\epsilon n\rceil}{J}.
\end{aligned} \tag{18}
$$

To proceed, one can write

$$
\begin{aligned}
\Pr(\bar{X}_{j,i} \leq -a) &= \Pr\left( \frac{\bar{X}_{j,i}}{\sqrt{\text{Var}(X)/B}} \leq -\frac{a}{\sqrt{\text{Var}(X)/B}} \right) \\
&\overset{(a)}{\leq} \Phi\left( -\frac{a}{\sqrt{\text{Var}(X)/B}} \right) + \frac{0.5\rho}{\sigma^3\sqrt{B}} \\
&\overset{(b)}{\leq} \exp\left\{ -\frac{Ba^2}{2\text{Var}(X)} \right\} + \frac{0.5\rho}{\sigma^3\sqrt{B}} \\
&\leq \exp\left\{ -\frac{Ba^2}{2\sigma^2} \right\} + \frac{0.5\rho}{\sigma^3\sqrt{B}}.
\end{aligned} \tag{19}
$$

Here, $(a)$ follows from the Berry-Esseen bound (Lemma 5). In $(b)$, we use the concentration inequality for standard Gaussian distribution. Combining the above inequalities and recalling our choices of $J$, $B$, and $n$, we conclude that, with probability at least $1 - \delta$ and for all $1 \leq i \leq d$,

$$
\begin{aligned}
\frac{1}{J} \sum_{j=1}^{J} \widetilde{\text{sign}} \left( \bar{X}_{j,i} + a \right) &\geq \frac{|\mathcal{J}_{\text{clean}}|}{J} \left( 1 - 2 \left( \exp\left\{ -\frac{Ba^2}{2\sigma^2} \right\} + \frac{0.5\rho}{\sigma^3 \sqrt{B}} \right) - \sqrt{\frac{2 \log(2d/\delta)}{|\mathcal{J}_{\text{clean}}|}} \right) - \frac{\lceil \epsilon n \rceil}{J} \\
&\geq 0.99 \cdot \left( 1 - 2 \cdot \left( e^{-2} + 0.025 \right) - \sqrt{\frac{100}{99} \cdot \frac{1}{1 \times 10^6}} \right) - 0.01 \\
&\geq \frac{3}{5}.
\end{aligned}
\tag{20}
$$

This completes the proof of the first statement.

_Case 2_: $0 \leq a \leq 0.001\sigma\sqrt{\epsilon}$. In this case, it suffices to provide an upper bound for $\left| \frac{1}{J} \sum_{j=1}^{J} \widetilde{\text{sign}} \left( \bar{X}_{j,i} + a \right) \right|$. Following a similar derivation as in _Case 1_, with probability at least $1 - \delta$ and for all $1 \leq i \leq d$, we have

$$
\begin{aligned}
&\left| \frac{1}{J} \sum_{j=1}^{J} \widetilde{\text{sign}} \left( \bar{X}_{j,i} + a \right) \right| \\
&\leq \frac{|\mathcal{J}_{\text{clean}}|}{J} \left( 1 - 2\Phi \left( -\frac{a}{\sqrt{\text{Var}(X)/B}} \right) + \frac{\rho}{\sigma^3 \sqrt{B}} + \sqrt{\frac{2 \log(2d/\delta)}{|\mathcal{J}_{\text{clean}}|}} \right) + \frac{\lceil \epsilon n \rceil}{J} \\
&= \frac{|\mathcal{J}_{\text{clean}}|}{J} \left( 2\Phi(0) - 2\Phi \left( -\frac{a}{\sqrt{\text{Var}(X)/B}} \right) + \frac{\rho}{\sigma^3 \sqrt{B}} + \sqrt{\frac{2 \log(2d/\delta)}{|\mathcal{J}_{\text{clean}}|}} \right) + \frac{\lceil \epsilon n \rceil}{J} \\
&\overset{(a)}{\leq} \frac{1.98a}{\sqrt{\sigma^2/B}} + \frac{0.99\rho}{\sigma^3 \sqrt{B}} + \sqrt{\frac{1.98 \log(2d/\delta)}{J}} + \frac{\lceil \epsilon n \rceil}{J} \\
&\leq 1.98\sqrt{B}\epsilon + 0.05 + \sqrt{\frac{1.98}{9000}} + 0.01 \\
&\leq 0.08.
\end{aligned}
\tag{21}
$$

Here, in $(a)$, we use the anti-concentration for the standard Gaussian distribution. This completes the proof of the second statement. $\square$

We are now ready to present the proof of Theorem 1. To this goal, we first present a more precise version of its statement.

**Theorem 4** (Convergence guarantee for SubGM)**.** _Let $\mathbb{P}$ be a distribution on $\mathbb{R}^d$ with an unknown $k$-sparse mean $\boldsymbol{\mu}^\star$, unknown covariance matrix $\boldsymbol{\Sigma} \preceq \sigma^2 \boldsymbol{I}$, and unknown coordinate-wise third moment satisfying $\mathbb{E}[|X_i - \mu_i^\star|^3] \leq 0.005\sigma^3/\sqrt{\epsilon}, \forall 1 \leq i \leq d$. Suppose a sample set of size $n \geq 20000 \log(2d/\delta)/\epsilon$ is collected according to the strong contamination model (Definition 1) with corruption parameter $\epsilon$. Upon setting the stepsize $\eta \leq \sigma\sqrt{\epsilon}/\mu_{\max}^\star$ and the initialization scale $0 < \alpha \leq 0.001\sigma\sqrt{\epsilon/d} \wedge \mu_{\max}^{\star -5}$ in Algorithm 1, with a probability of at least $1 - \delta$, the following statements hold for any iteration $\frac{2}{\eta} \log(1/\alpha) \leq T \leq \frac{6}{\eta} \log(1/\alpha)$:_

- **_Near optimal $\ell_2$-error._** _The $\ell_2$-error is upper-bounded by_

$$
\|\hat{\boldsymbol{\mu}}(T) - \boldsymbol{\mu}^\star\| \leq 31\sigma\sqrt{k}\epsilon.
\tag{22}
$$

- **_Coordinate-wise error bound._** _We obtain_

$$
\begin{aligned}
|\hat{\mu}_i(T) - \mu_i^\star| &\leq 30\sigma\sqrt{\epsilon}, &\text{where } \mu_i^\star \neq 0, \\
|\hat{\mu}_i(T)| &\leq \alpha, &\text{where } \mu_i^\star = 0.
\end{aligned}
\tag{23}
$$

Before proceeding to the proof, we note that the second statement of Theorem 4 together with the assumption $\epsilon \leq \mu_{\min}^{\star 2}/(961\sigma^2)$ readily implies $\hat{\mu}_i(T) \geq \sigma\sqrt{\epsilon}$ for every $i$ such that $\mu_i^{\star} \neq 0$, leading to the second statement of Theorem 1.

*Proof of Theorem 4.* Let us define $\mathcal{I}_{\text{residual}} = \{i : \mu_i^{\star} = 0\}$ and $\mathcal{I}_{\text{signal}} = \{i : \mu_i^{\star} \neq 0\}$. We analyze coordinate-wise dynamics $\hat{\mu}_i(t) := u_i^2(t) - v_i^2(t)$ separately for signals $\mathcal{I}_{\text{signal}}$ and residuals $\mathcal{I}_{\text{residual}}$.

**Signal dynamics.** Without loss of generality, we assume that $\mu_i^{\star} > 0$. Let us first revisit the update rule for SubGM:

$$
\begin{aligned}
u_i(t+1) &= \left(1 + \eta\frac{1}{J}\sum_{j=1}^{J}\widetilde{\text{sign}}\left(\bar{X}_{j,i} - \hat{\mu}_i(t)\right)\right)u_i(t), \\
v_i(t+1) &= \left(1 - \eta\frac{1}{J}\sum_{j=1}^{J}\widetilde{\text{sign}}\left(\bar{X}_{j,i} - \hat{\mu}_i(t)\right)\right)v_i(t).
\end{aligned}
\tag{24}
$$

We further divide our analysis into two cases depending on the magnitude of $|\mu_i^{\star}|$.

*Case 1*: $\mu_i^{\star} \geq 20\sigma\sqrt{\epsilon}$. We define $T_i = \{\min t : \mu_i^{\star} - \hat{\mu}_i(t) < 20\sigma\sqrt{\epsilon}\}$. Hence, for all $0 \leq t \leq T_i$, the first statement of Lemma 6 can be invoked to show

$$
0.6 \leq \frac{1}{J}\sum_{j=1}^{J}\widetilde{\text{sign}}\left(\bar{X}_{j,i} - \hat{\mu}_i(t)\right) \leq 1.
\tag{25}
$$

By incorporating this into Equation (24), we obtain

$$
\begin{aligned}
u_i^2(t+1) &\geq (1 + 0.6\eta)^2 \, u_i^2(t) \geq (1 + 1.2\eta)u_i^2(t), \tag{26} \\
v_i^2(t+1) &\leq (1 - 0.6\eta)^2 \, v_i^2(t) \leq v_i^2(t). \tag{27}
\end{aligned}
$$

Notice that $v_i(0) = \alpha$ at the initialization. We find that $v_i^2(t) \leq \alpha^2, \forall 0 \leq t \leq T_i$, which remains adequately small throughout the trajectory. Next, we examine the dynamics of $u_i^2(t)$. Taking into account that $u_i^2(0) = \alpha^2$ and $u_i^2(t) \geq (1 + 1.2\eta)^t u_i^2(0)$, we have that within $T_i \leq \frac{5}{3\eta}\log\left(\frac{|\mu_i^{\star}|}{\alpha}\right)$ iterations, the following holds

$$
u_i^2(T_i) \geq \alpha^2(1 + 1.2\eta)^{T_i} \geq \mu_i^{\star} - 10\sigma\sqrt{\epsilon}.
\tag{28}
$$

This implies

$$
\hat{\mu}_i(T_i) = u_i^2(T_i) - v_i^2(T_i) \geq \mu_i^{\star} - 10\sigma\sqrt{\epsilon} - \alpha^2 \geq \mu_i^{\star} - 20\sigma\sqrt{\epsilon},
\tag{29}
$$

since $v_i^2(T_i) \leq \alpha^2$. Next, we show that $|\mu_i^{\star} - \hat{\mu}_i(T_i)| \leq 20\sigma\sqrt{\epsilon}$. To this goal, when $t < T_i$, we provide an upper bound on the difference between two consecutive iterations as follows

$$
\begin{aligned}
|\hat{\mu}_i(t+1) - \hat{\mu}_i(t)| &\leq \left|u_i^2(t+1) - u_i^2(t)\right| + \left|v_i^2(t+1) - v_i^2(t)\right| \\
&\overset{(a)}{\leq} \left|\left(1 + \eta\frac{1}{J}\sum_{j=1}^{J}\widetilde{\text{sign}}\left(\bar{X}_{j,i} - \hat{\mu}_i(t)\right)\right)^2 - 1\right| \cdot u_i^2(t) + \alpha^2 \\
&\overset{(b)}{\leq} \left((1 + \eta)^2 - 1\right) u_i^2(t) + \alpha^2 \\
&\overset{(c)}{\leq} 3\eta\mu_i^{\star} + 2\alpha^2 \\
&\overset{(d)}{\leq} 4\sigma\sqrt{\epsilon}.
\end{aligned}
\tag{30}
$$

Here in $(a)$, we use the fact that $\left|v_i^2(t+1) - v_i^2(t)\right| \leq \max\{v_i^2(t+1), v_i^2(t)\} \leq \alpha^2$. In $(b)$, we use the estimate $0.6 \leq \frac{1}{J}\sum_{j=1}^{J}\widetilde{\text{sign}}\left(\bar{X}_{j,i} - \hat{\mu}_i(t)\right) \leq 1$. In $(c)$, we use the fact that $u_i^2(t) \leq \hat{\mu}_i(t) + v_i^2(t) \leq \mu_i^{\star} + \alpha^2$. Lastly, in $(d)$, we use the condition that $\eta \leq \frac{\sigma\sqrt{\epsilon}}{\mu_{\max}^{\star}}$ and $\alpha^2 \leq 0.5\sigma\sqrt{\epsilon}$. Hence, we have

$$
\hat{\mu}_i(T_i) - \mu_i^{\star} \leq \underbrace{\hat{\mu}_i(T_i - 1) - \mu_i^{\star}}_{\leq 0 \text{ by definition of } T_i} + (\hat{\mu}_i(T_i) - \hat{\mu}_i(T_i - 1)) \leq 4\sigma\sqrt{\epsilon}.
\tag{31}
$$

Combining with the fact that $\hat{\mu}_i(T_i) - \mu_i^\star \geq -20\sigma\sqrt{\epsilon}$, we derive that $|\mu_i^\star - \hat{\mu}_i(T_i)| \leq 20\sigma\sqrt{\epsilon}$.

We will now demonstrate that for any $t \geq T_i$, the condition $|\mu_i^\star - \hat{\mu}_i(t)| \leq 30\sigma\sqrt{\epsilon}$ always holds. Using the fact $n \geq 20000 \log(2d/\delta)/\epsilon$ and Theorem 5 (in the appendix), we have $|\hat{\mu}_{\mathsf{MoM}} - \mu_i^\star| \leq 5\sigma\sqrt{\epsilon}$. Then, the triangle inequality implies

$$|\mu_i^\star - \hat{\mu}_i(t)| \leq |\hat{\mu}_{\mathsf{MoM}} - \mu_i^\star| + |\hat{\mu}_{\mathsf{MoM}} - \hat{\mu}_i(t)|. \tag{32}$$

Therefore, it suffices to show that $|\hat{\mu}_{\mathsf{MoM}} - \hat{\mu}_i(t)| \leq 25\sigma\sqrt{\epsilon}$ for every $t \geq T_i$. To this goal, we use induction on $t$. For $t = T_i$, we have

$$|\hat{\mu}_{\mathsf{MoM}} - \hat{\mu}_i(T_i)| \leq |\hat{\mu}_{\mathsf{MoM}} - \mu_i^\star| + |\mu_i^\star - \hat{\mu}_i(T_i)| \leq 25\sigma\sqrt{\epsilon}. \tag{33}$$

Now, let us assume that at time $t \geq T_i$, $|\hat{\mu}_{\mathsf{MoM}} - \hat{\mu}_i(t)| \leq 25\sigma\sqrt{\epsilon}$. Without loss of generality, we assume $\hat{\mu}_i(t) < \hat{\mu}_{\mathsf{MoM}}$. Based on the definition of the $\mathsf{MoM}$ estimator, we have

$$\sum_{j=1}^{J} \widetilde{\mathrm{sign}}\left(\bar{X}_{j,i} - \hat{\mu}_i(t)\right) \geq 0. \tag{34}$$

Let $\beta_i(t) = \frac{1}{J}\sum_{j=1}^{J} \widetilde{\mathrm{sign}}\left(\bar{X}_{j,i} - \hat{\mu}_i(t)\right)$. With this notation, we can derive the following inequality

$$\hat{\mu}_i(t+1) - \hat{\mu}_i(t) = (2\eta\beta_i(t) + \eta^2\beta_i^2(t))u_i^2(t) + (2\eta\beta_i(t) - \eta^2\beta_i^2(t))v_i^2(t) \geq 0, \tag{35}$$

where in the last inequality, we use the fact that $u_i^2(t) \geq v_i^2(t)$ and $\beta_i(t) \geq 0$. On the other hand, following exactly the same argument in Equation (30), we have

$$\hat{\mu}_i(t+1) - \hat{\mu}_i(t) \leq 4\sigma\sqrt{\epsilon}. \tag{36}$$

By combining the above two inequalities, we establish that $|\hat{\mu}_{\mathsf{MoM}} - \hat{\mu}_i(t+1)| \leq 25\sigma\sqrt{\epsilon}$. This completes the proof of induction.

_Case 2_: $|\mu_i^\star| \leq 20\sigma\sqrt{\epsilon}$. Since $\hat{\mu}_i(0) = 0$, at iteration $t = 0$ we already have $|\mu_i^\star - \hat{\mu}_i(t)| \leq 20\sigma\sqrt{\epsilon}$. Consequently, the analysis reduces to the last phase of _Case 1_, from which we can conclude $|\mu_i^\star - \hat{\mu}_i(t)| \leq 30\sigma\sqrt{\epsilon}$ for all $t \geq 0$.

**Residual dynamics.** In this case, we employ induction on $t$ to demonstrate that $|u_i^2(t) - v_i^2(t)| \leq \alpha$ for all $0 \leq t \leq T$. For the base case, this relationship is valid as $u_i^2(0) - v_i^2(0) = 0$. Assuming that this relation holds at time $t$, we can refer to Lemma 6 and deduce

$$-0.08 \leq \frac{1}{J}\sum_{j=1}^{J} \widetilde{\mathrm{sign}}\left(\bar{X}_{j,i} - \hat{\mu}_i(t)\right) \leq 0.08. \tag{37}$$

Hence, we have

$$\begin{aligned}
u_i^2(t+1) &\leq (1+0.08\eta)^2 u_i^2(t) \leq (1+\eta/6)\, u_i^2(t), \\
v_i^2(t+1) &\leq (1+0.08\eta)^2 v_i^2(t) \leq (1+\eta/6)\, v_i^2(t).
\end{aligned} \tag{38}$$

Therefore, for all $t \leq \frac{6}{\eta}\log\left(\frac{1}{\alpha}\right)$, we obtain

$$\left|u_i^2(t) - v_i^2(t)\right| \leq \max\left\{u_i^2(t), v_i^2(t)\right\} \leq \alpha^2(1+\eta/6)^t \leq \alpha. \tag{39}$$

**Putting everything together.** Finally, since we set $\alpha \leq \frac{0.001}{\sqrt{d}}\sigma\sqrt{\epsilon} \wedge \mu_{\max}^{\star-5}$, for any $\frac{2}{\eta}\log\left(\frac{1}{\alpha}\right) \leq T \leq \frac{6}{\eta}\log\left(\frac{1}{\alpha}\right)$, we have

$$\|\hat{\boldsymbol{\mu}}(T) - \boldsymbol{\mu}^\star\|_2 \leq \sqrt{k} \cdot 30\sigma\sqrt{\epsilon} + \sqrt{d}\alpha \leq 31\sigma\sqrt{k\epsilon}. \tag{40}$$

This completes the proof. $\qquad\square$

### 5.3 Proof of Theorem 2

The proof follows by combining Theorem 1 and Proposition 2. First, for the data distribution and corruption model considered in Theorem 1, once we set the sample size $n \gtrsim \log(d/\delta)/\epsilon$, then with probability at least $1 - \delta/2$, we can successfully determine the location of the top-$k$ nonzero elements. For brevity, we represent the indices of these top-$k$ elements as $I_k$. Following the successful determination of these indices, we can then narrow our focus to a $k$-dimensional subproblem on the dataset $S_k := \{[X_i]_{I_k} : X_i \in S\}$ with the mean $[\mu^\star]_{I_k}$. We can then apply Proposition 2 to this reduced dataset. Specifically, once the sample size satisfies $n \gtrsim (k + \log(d/\delta))/\epsilon$, there exists an estimator such that with probability at least $1 - \delta/2$, it can output a $\hat{\mu}$ satisfying $\|\hat{\mu} - [\mu^\star]_{I_k}\| \lesssim \sigma\sqrt{\epsilon}$.

Combining these two steps via a simple union bound, we know that with a probability of at least $1 - \delta$, our two-stage estimator $\hat{\mu}$ satisfies $\|\hat{\mu} - \mu^\star\| \lesssim \sigma\sqrt{\epsilon}$. This concludes the proof. $\qquad\square$

### 5.4 Proof of Theorem 3

Consider two probability distributions $\mathbb{P}_1$ and $\mathbb{P}_2$, where $\mathbb{P}_2 = (1-\epsilon)\mathbb{P}_1 + \epsilon\mathbb{Q}$ for some distribution $\mathbb{Q}$. Suppose we draw $n$ i.i.d. samples from $\mathbb{P}_1$. Under the strong contamination model (Definition 1) with parameter $\epsilon$, this same set of samples can be equivalently viewed as $\epsilon$-corrupted samples from $\mathbb{P}_2$. Consequently, no algorithm can distinguish between the two cases (see Li (2019) for details).

Therefore, it suffices to construct two probability distributions satisfying the conditions in Theorem 3. Without loss of generality, we focus on the one-dimensional case, since additional coordinates can be set identically. We require two distributions $\mathbb{P}_1, \mathbb{P}_2$ such that:

- Both distributions have variance at most $\sigma^2$ and third central moment at most $\sigma^3/\sqrt{\epsilon}$;

- $\mathbb{P}_2$ can be written as $(1 - \epsilon)\mathbb{P}_1 + \epsilon\mathbb{Q}$ for some distribution $\mathbb{Q}$;

- Their means $\mu_1, \mu_2$ satisfy $|\mu_1 - \mu_2| \geq \sigma\sqrt{\epsilon}$.

Following Li (2019), we construct $\mathbb{P}_1$ as the point mass at 0, and let $\mathbb{P}_2 = (1 - \epsilon)\mathbb{P}_1 + \epsilon\mathbb{Q}$, where $\mathbb{Q}$ is the point mass at $\sigma/\sqrt{\epsilon}$. It is straightforward to verify that $\mathbb{P}_1$ and $\mathbb{P}_2$ satisfy all three conditions, completing the proof. $\qquad\square$

## 6 Simulation

In this section, we present numerical simulations to corroborate the theoretical results established in Section 4. Further implementation details, together with additional simulation studies, are deferred to the appendix. The complete codebase is publicly accessible at `https://github.com/ying-hui-he/Robust_mean_estimation`.

**Simulation setup.** All the experiments are conducted on a MacBook Pro 2021 with the Apple M1 Pro chip and a 16GB unified memory. We pick three representative heavy-tailed probability distributions: Fisk, Pareto, and Student's $t$. To make a fair comparison, we fix the data dimension at $d = 100$ and use the constant-bias noise model introduced in Cheng et al. (2021) to generate outliers. Unless otherwise stated, we set the corruption ratio at $\epsilon = 0.1$ and the sparsity level at $k = 4$. As for the algorithm in Stage 2, we utilize the filter-based algorithm `RME_sp` introduced in Diakonikolas et al. (2019b). Furthermore, we compare our algorithms with the *oracle* estimator, which applies truncated coordinate-wise MoM to the clean data: we estimate each coordinate via MoM, retain the top-$k$ entries, and set the remaining coordinates to zero. In all of our simulations, we set the number of iterations of SubGM to 200, which is in line with our theoretical results.

**Identification of top-$k$ elements.** In this experiment, we evaluate the success rate under varying corruption ratios $\epsilon$, while keeping all other parameters fixed. Our theoretical result (Theorem 1) indicates that

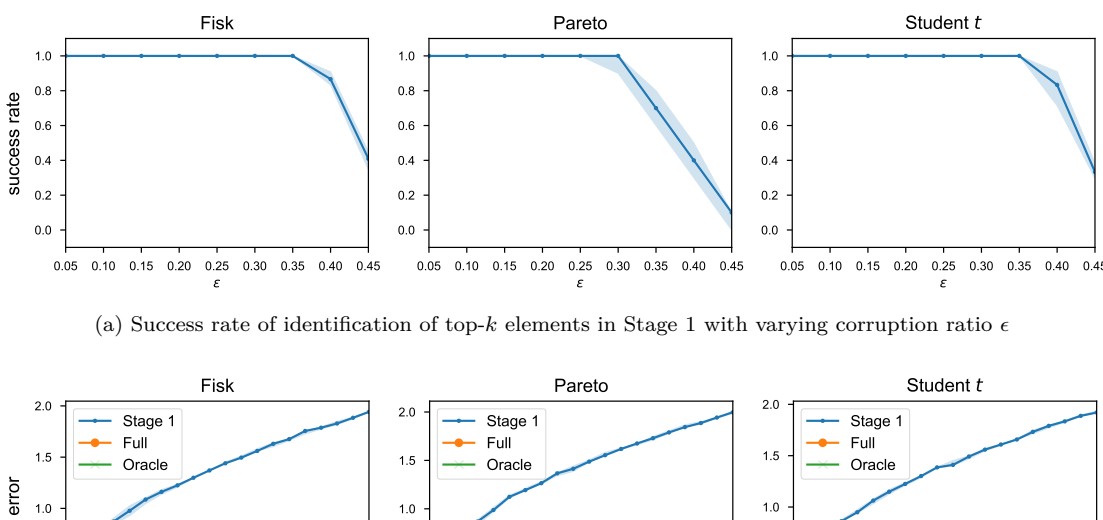

(a) Success rate of identification of top-$k$ elements in Stage 1 with varying corruption ratio $\epsilon$

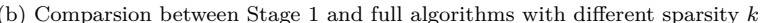

(b) Comparsion between Stage 1 and full algorithms with different sparsity $k$

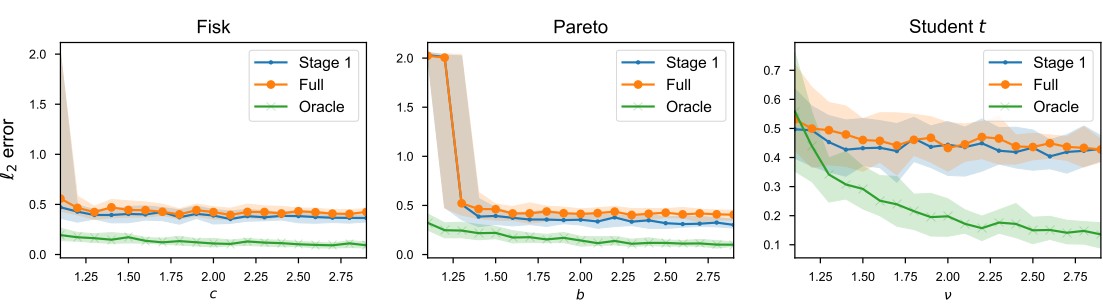

(c) Performances of Stage 1 and full algorithms in infinite variance regime

Figure 2: The data dimension is fixed at $d = 100$. Unless otherwise specified, the corruption ratio is set to $\epsilon = 0.1$ and the sparsity level to $k = 4$. For the first two simulations, the distribution parameters $c, b, \nu$ are set to 3.1 (see Appendix B for further details). The sample size is $n = 600$ in the first and third simulations, while in the second simulation it is scaled as $n = 100k$, varying proportionally with the sparsity level.

provable identification is possible only when $\epsilon \lesssim \mu_{\min}^{\star 2}/\sigma^2$, suggesting that the success rate should deteriorate as $\epsilon$ increases. We define the recovered index set obtained by SubGM as $I$, and the true index set of the top-$k$ elements as $I_k$. The success rate is then measured as $|I \cap I_k|/|I \cup I_k|$. The results, presented in Figure 2a, are averaged over 50 independent trials for each setting. Notably, SubGM achieves exact recovery of the true index set $I$ even when up to 30% of the samples are corrupted, highlighting the robustness and practical effectiveness of our method.

**Comparison between Stage 1 and full algorithms.** We evaluate the $\ell_2$-error of the Stage 1 and full algorithms across varying sparsity levels $k$. Our theoretical results predict a gap in $\ell_2$-error between the two algorithms—$\mathcal{O}(\sigma\sqrt{k\epsilon})$ versus $\mathcal{O}(\sigma\sqrt{\epsilon})$—when $k$ is sufficiently large. To minimize the influence of sample size, we set $n = 100k$, ensuring a sufficiently large number of samples. As shown in Figure 2b, the two algorithms perform comparably when $k$ is small. However, as $k$ increases, the $\ell_2$-error of Stage 1 grows sublinearly, while the full algorithm maintains a stable error level. These empirical findings are fully consistent with our theoretical predictions.

**Infinite variance regime.** In this experiment, we evaluate the performance of our algorithm in the infinite variance regime, fixing the sparsity level at $k = 4$ and the sample size at $n = 600$. When the distribution parameters $c, b, \nu$ fall within the interval $(1, 2]$, the Fisk, Pareto, and Student's $t$ distributions all exhibit infinite variance (see Appendix B for further details). As shown in Figure 2c, both Stage 1 and the full algorithm maintain strong performance in this setting, suggesting that our theoretical guarantees may extend to the infinite variance regime. Notably, Stage 1 consistently outperforms the full algorithm across all three distributions, implying that SubGM may possess greater robustness than existing estimators under infinite variance.

## 7 Conclusion and Future Directions

Many estimation tasks in statistics become notoriously difficult in the robust setting when certain assumptions on the data are lifted. For instance, almost all statistically optimal robust mean estimators suffer from overwhelmingly high computational costs. While classical results in robust statistics have shed light on the statistical limits of robust estimation, its computational aspects have mostly remained elusive. In this work, we aim to bridge this gap by presenting the *first* computationally efficient and statistically optimal method for robust sparse mean estimation, thereby overcoming a conjectured computational-statistical barrier under moderate conditions. We believe that our method can be extended to other estimation tasks in robust statistics. In the following discussion, we highlight two promising directions for future research.

**Beyond bounded variance.** We conjecture that our method can be applied to distributions with unbounded variance. Our current theoretical result requires a bounded variance and a coordinate-wise $\epsilon$-dependent bounded third moment. Nonetheless, we speculate that these conditions could be relaxed with a more delicate analysis. Indeed, our simulations on the data drawn from the distributions with unbounded variance, namely the two-sided Pareto distribution and Student's $t$ distribution, strongly suggest that our estimator can achieve desirable performance even when the variance is infinite.

**Beyond robust mean estimation.** Another direction is to extend our approach to other robust estimation tasks, including robust PCA (Diakonikolas et al., 2019b), robust covariance estimation (Cheng et al., 2019b), and robust linear regression (Diakonikolas et al., 2019c). Our method crucially relies on incremental learning which is ubiquitous in diverse settings, spanning from linear regimes (such as linear regression (Ma & Fattahi, 2022), matrix factorization (Li et al., 2020), and tensor factorization (Razin et al., 2021; 2022; Ma et al., 2022)) to nonlinear regimes (like neural networks (Frei et al., 2022)). We are optimistic that our techniques can be used to design statistical and computationally efficient algorithms across a broader range of tasks in robust statistics.

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

## A  MoM Estimator under Strong Contamination Model

In this section, we prove the key properties of the 1-dimensional and high-dimensional MoM estimators under the strong contamination model (Definition 1). The following is a more precise statement of Proposition 1, which is adapted from Fact 2.1. in Diakonikolas et al. (2022b). As the complete proof does not appear in the original source, we include it here for completeness.

**Proposition 3** (One-dimensional MoM estimator). *Consider a corruption parameter $\epsilon \leq \frac{1}{8}$, failure probability $\delta$, and a set $S$ of $n$ many $\epsilon$-corrupted samples from a distribution $\mathbb{P}$ with mean $\mu^\star$ and variance $\mathbb{E}[(X - \mu^\star)^2] \leq \sigma^2$. Then, with probability at least $1 - \delta$, the MoM estimator $\hat{\mu}_{MoM}$ satisfies $|\hat{\mu}_{MoM} - \mu^\star| \leq \sigma \left( 4\sqrt{2} \left( \sqrt{\epsilon} + \sqrt{1/n} \right) + 16\sqrt{\log(1/\delta)/n} \right)$.*

*Proof.* We partition the index set of the subgroups $\{1, \cdots, J\}$ into two parts: $\mathcal{J}_{\text{clean}}$ and $\mathcal{J}_{\text{outlier}}$. Here $\mathcal{J}_{\text{clean}}$ comprises all the subgroups without outliers, and $\mathcal{J}_{\text{outlier}}$ consists of subgroups containing at least one outlier. According to our strong contamination model, we have $|\mathcal{J}_{\text{outlier}}| \leq \lceil \epsilon n \rceil$. Subsequently, we observe that

$$\{|\hat{\mu}_{\text{MoM}} - \mu^\star| \geq \xi\} \subseteq \left\{ \sum_{j \in \mathcal{J}_{\text{clean}}} \mathbb{I}(|\bar{X}_j - \mu^\star| \geq \xi) \geq \frac{J}{2} - \lceil \epsilon n \rceil \right\}. \tag{41}$$

Here, $\bar{X}_j = \frac{1}{B} \sum_{i \in S_j} X_i$, where $B = n/J$ is the size of each subgroup and $S_j$ is the $j$-th subgroup. For simplicity, let us denote $Z_j = \mathbb{I}(|\bar{X}_j - \mu^\star| \geq \xi)$ and $p_\xi = \Pr\left(|\bar{X}_j - \mu^\star| \geq \xi\right)$. Then, the above inclusion implies

$$\begin{aligned} \Pr\left(|\hat{\mu}_{\text{MoM}} - \mu^\star| \geq \xi\right) &\leq \Pr\left( \sum_{j \in \mathcal{J}_{\text{clean}}} Z_j \geq \frac{J}{2} - \lceil \epsilon n \rceil \right) \\ &= \Pr\left( \frac{1}{|\mathcal{J}_{\text{clean}}|} \sum_{j \in \mathcal{J}_{\text{clean}}} (Z_j - \mathbb{E}[Z_j]) \geq \frac{J/2 - \lceil \epsilon n \rceil}{|\mathcal{J}_{\text{clean}}|} - p_\xi \right). \end{aligned} \tag{42}$$

Since $Z_j$ is bounded, we can apply Hoeffding's inequality (Lemma 2) to obtain

$$\Pr\left(|\hat{\mu}_{\text{MoM}} - \mu^\star| \geq \xi\right) \leq \exp\left\{ -2|\mathcal{J}_{\text{clean}}| \left( \frac{J/2 - \lceil \epsilon n \rceil}{|\mathcal{J}_{\text{clean}}|} - p_\xi \right)^2 \right\}. \tag{43}$$

Moreover, we can use Chebyshev's inequality (Lemma 1) to establish an upper bound for $p_\xi$:

$$p_\xi = \Pr\left(|\bar{X}_j - \mu^\star| \geq \xi\right) \leq \frac{\sigma^2}{B\xi^2} = \frac{J\sigma^2}{n\xi^2}. \tag{44}$$

Upon defining $J = 4\lceil \epsilon n \rceil + 32\log(1/\delta)$ and $\xi = \sigma\left( 4\sqrt{2}\left( \sqrt{\epsilon} + \sqrt{1/n} \right) + 16\sqrt{\log(1/\delta)/n} \right)$, we have the following estimates

$$\begin{aligned} |\mathcal{J}_{\text{clean}}| &\geq J - \lceil \epsilon n \rceil \geq 32\log(1/\delta); \\ \frac{J/2 - \lceil \epsilon n \rceil}{|\mathcal{J}_{\text{clean}}|} &\geq \frac{J/2 - \lceil \epsilon n \rceil}{J} \geq \frac{1}{4}; \\ p_\xi &\leq \frac{J\sigma^2}{n\xi^2} \leq \frac{1}{8}. \end{aligned} \tag{45}$$

Combining these bounds with Equation (43), we obtain

$$\begin{aligned} &\Pr\left( |\hat{\mu}_{\text{MoM}} - \mu^\star| \geq \sigma\left( 4\sqrt{2}\left( \sqrt{\epsilon} + \sqrt{1/n} \right) + 16\sqrt{\log(1/\delta)/n} \right) \right) \\ &\leq \exp\left\{ -2 \cdot 32\log(1/\delta) \cdot \left( \frac{1}{4} - \frac{1}{8} \right)^2 \right\} = \delta. \end{aligned} \tag{46}$$

This completes the proof. □

Directly applying MoM estimator to each coordinate of a $d$-dimensional dataset leads to the following proposition.

**Theorem 5** (High dimensional coordinate-wise MoM estimator)**.** *Consider a corruption parameter $\epsilon$, failure probability $\delta$, and a set $S$ of $n$ many $\epsilon$-corrupted samples from a distribution $\mathbb{P}$ with mean $\mu^\star$ and coordinate-wise variance $\mathbb{E}[(X - \mu^\star)^2] \leq \sigma^2, \forall 1 \leq i \leq d$. Then, with probability at least $1 - \delta$, the coordinate-wise MoM estimator $\hat{\mu}_{MoM}$ satisfies $\|\hat{\boldsymbol{\mu}}_{MoM} - \boldsymbol{\mu}^\star\|_\infty \leq \sigma \left(4\sqrt{2}\left(\sqrt{\epsilon} + \sqrt{1/n}\right) + 16\sqrt{\log(d/\delta)/n}\right)$ and $\|\hat{\boldsymbol{\mu}}_{MoM} - \boldsymbol{\mu}^\star\|_2 \leq \sigma\sqrt{d}\left(4\sqrt{2}\left(\sqrt{\epsilon} + \sqrt{1/n}\right) + 16\sqrt{\log(d/\delta)/n}\right)$.*

*Proof.* The proof follows directly from Proposition 3 and a simple union bound. □

# B  Additional Simulations

## B.1  Experimental Details

We run our simulations on three heavy-tailed distributions: Fisk, Pareto, and Student's $t$ distributions. In each case, we apply a symmetrization trick to make the density function symmetric around zero. The density function of the Fisk distribution with parameter $c$ is expressed as follows:

$$f(x; c) = \frac{c|x|^{c-1}}{2(1 + |x|^c)^2} \quad \text{for } x \in \mathbb{R}, c > 0. \tag{47}$$

The density function of the Pareto distribution with parameters $b$ is

$$f(x; b) = \left\{ \begin{array}{ll} \frac{b}{2|x|^{b+1}} & \text{for} \quad |x| \geq 1, \\ 0 & \text{for} \quad |x| < 1. \end{array} \right., \quad \text{for } x \in \mathbb{R}, b > 0. \tag{48}$$

Lastly, the density function for student $t$-distribution is

$$f(x; \nu) = \frac{\Gamma\left(\frac{\nu+1}{2}\right)}{\sqrt{\nu\pi}\Gamma(\nu/2)} \left(1 + \frac{x^2}{\nu}\right)^{-(\nu+1)/2} \quad \text{for } x \in \mathbb{R}, \nu > 0. \tag{49}$$

Here $\Gamma$ is the gamma function. In all three distributions described above, the parameters $c, b, \nu$ correspondingly denote the existence of the $c, b, \nu$-th moment. For instance, when $c, b, \nu$ fall within the range of $(1, 2]$, the variances are infinite. Regarding the outliers, we generate them via the constant-bias noise model as introduced in Cheng et al. (2021).

Furthermore, unless stated otherwise, all simulations are conducted with the following predefined settings: data dimension $d$ is set to 100, sparsity level $k$ is set to 4 with nonzero elements being $[10, -5, -4, 2]$, sample size $m$ is set to 600, and the corruption ratio $\epsilon$ is set at 10%. As for our algorithm, we set the number of subgroups to be $J = 1.5\lceil\epsilon n\rceil + 150$. Note that, compared to the theoretical choice of $J = 100\lceil\epsilon n\rceil$ in Algorithm 1, we choose a smaller $J$ to make our algorithm work for a larger corruption ratio $\epsilon$ in practice. Moreover, in SubGM, we set the initialization scale $\alpha = 10^{-5}$ and the step-size $\eta = 0.05$. We note that these choices of parameters differ from those used in the experiments illustrated in Figure 1.

We select `sparse_GD` (Cheng et al., 2021) and `sparse_filter` (Diakonikolas et al., 2019b) as our benchmark algorithms. We note that these algorithms *do not* come with theoretical assurances in the heavy-tailed setting. Nonetheless, we have empirically found that these two algorithms surpass others in performance, even in the heavy-tailed setting. We also highlight that the polynomial-time algorithms that come equipped with theoretical guarantees for the heavy-tailed setting (Diakonikolas et al., 2022b;a) are impractical since they rely on time-consuming methods such as sum-of-squares and ellipsoid methods.

We employ both `sparse_GD` and `sparse_filter` in the second stage of our algorithm, setting the sparsity parameter to $k = |I|$, where $I$ is the index set identified in the first stage. In total, we evaluate six

estimators: `oracle` (which removes all outliers and applies truncated coordinate-wise MoM to the clean data), `sparse_GD`, `sparse_filter`, `stage_1`, `full_GD` (our algorithm with `sparse_GD` in the second stage), and `full_filter` (our algorithm with `sparse_filter` in the second stage). In `stage_1`, we run SubGM for $T = 600$ iterations, whereas in `full_GD` and `full_filter`, we reduce the iteration count to $T = 200$ to lower computational cost.

## B.2   Sensitivity to Prior Knowledge of $k$

We underscore the fact that prior algorithms necessitate prior knowledge of the exact sparsity level $k$. In contrast, our approach can identify the sparsity level automatically. For this simulation, we assign a true sparsity level of $k = 10$ with nonzero components $[2, 2, 2, 2, 2, -2, -2, -2, -2, -2]$ and assess the performance of the benchmark algorithms, namely `sparse_GD` and `sparse_filter`, while varying the input $k'$, which is an upper bound of $k$, within the range of $[10, 40]$. As illustrated in Figure 3, the performance of these benchmark algorithms is highly sensitive to the choice of $k'$ across all examined distributions. Their performances further destabilize when the underlying distributions start to exhibit heavier tails. In contrast, our algorithm automatically recognizes the sparsity pattern across all scenarios. For all subsequent simulations, we provide the benchmark algorithms with the true sparsity level $k$ to ensure a fair comparison.

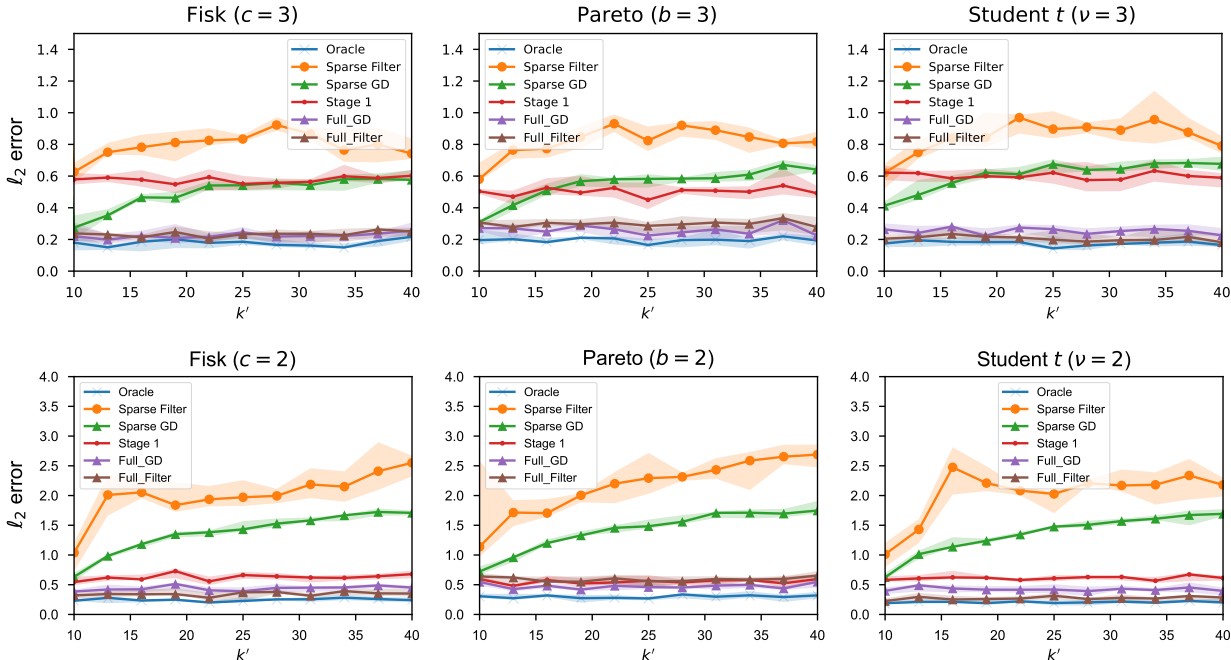

Figure 3: Comparison among different algorithms with varying input $k'$, where $k = 10$ and $k' \geq k$ is an upper bound of $k$. The second row corresponds to distributions with infinite variance.

## B.3   Performance with Different $k$

In this simulation, we evaluate the performance of various algorithms under different sparsity levels $k$. We set all nonzero entries of $\mu^\star$ to 2. As shown in the first row of Figure 4, all algorithms—except `stage_1` (as predicted by Theorem 1) and `sparse_filter` (which underperforms at larger sparsity levels $k$)—achieve $\ell_2$-error that remains largely independent of sparsity. In more heavy-tailed settings, depicted in the second row of Figure 4, all algorithms display an increase in $\ell_2$-error as $k$ grows. Nevertheless, across nearly all scenarios, our full algorithms (`full_GD` and `full_filter`) outperform the benchmarks. We further hypothesize that the weaker performance of `full_filter` for the Pareto distribution with $b = 2$ arises from the suboptimal performance of `sparse_GD` and `sparse_filter` when used in Stage 2.

**Clarification and connection to other figures.** While `sparse_GD` and `sparse_filter` appear to perform reasonably well in Figure 4 for moderate sparsity levels, their performance is in fact sensitive to the choice of the sparsity parameter $k'$. This sensitivity is explicitly illustrated in Figure 3, where even mild overestimation of $k$ leads to noticeable degradation. Moreover, as shown in Figure 5, this degradation becomes more pronounced under heavier-tailed distributions. Since `full_filter` relies on these methods in Stage 2, such sensitivity propagates to the full algorithm and explains its weaker performance in certain heavy-tailed settings.

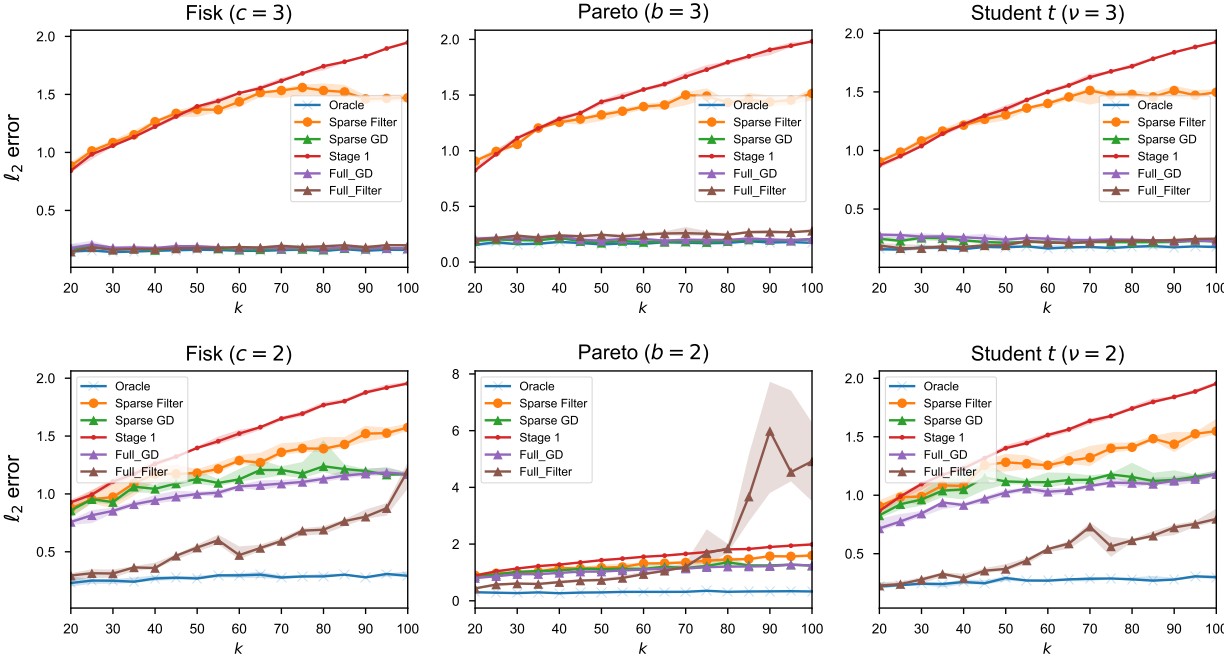

Figure 4: Comparison among different algorithms for varying sparsity levels $k$. The second row corresponds to distributions with infinite variance.

## B.4 Infinite Variance Regime

In this simulation, we evaluate the performance of the algorithms with respect to the heaviness of the tail distributions. As shown in Figure 5, we vary the parameters $c, b, \nu$ over the range $[1, 3.5]$. Smaller parameter values correspond to heavier tails, with values in the interval $(1, 2]$ resulting in distributions of infinite variance. Our algorithms (`stage_1`, `full_GD`, and `full_filter`) demonstrate superior robustness under these heavy-tailed conditions, highlighting the advantage of our approach.

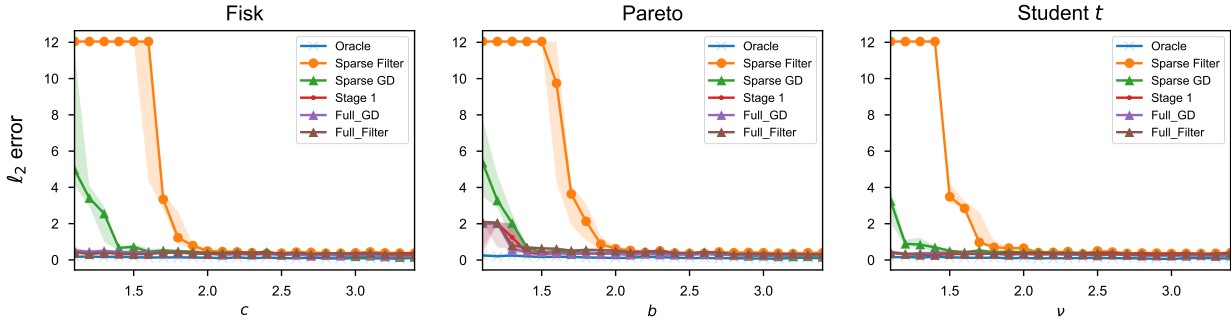

Figure 5: Comparison among different algorithms in the infinite variance regime.

### B.5 Performance with Different $\epsilon$

In this simulation, we study the relationship between the $\ell_2$-error and the corruption ratio $\epsilon$ across all six estimators. As shown in Figure 6, apart from the `Oracle`—whose error remains unaffected by $\epsilon$ (as expected)—our proposed algorithms (either single-stage or full version) consistently outperform the alternatives. While our theoretical analysis predicts an $\ell_2$-error of order $\mathcal{O}(\sqrt{\epsilon})$, the empirical results reveal an approximately linear dependence on $\epsilon$. We attribute this discrepancy to the non-adversarial nature of the outlier model used in our experiments. A promising direction for future work is to examine the performance of the proposed methods under truly adversarial corruptions. We believe that constructing such adversarial examples is far from trivial (see Shafieezadeh-Abadeh et al. (2023)) and is an interesting problem in its own right.

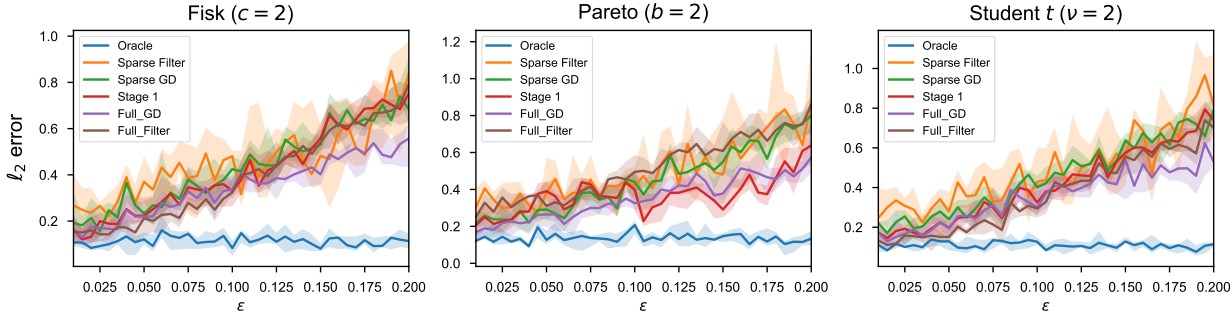

Figure 6: Comparison among different algorithms for different corruption rates $\epsilon$.

### B.6 Running Time

Next, we examine the running time of our algorithms and compare them with several relevant baseline methods. Specifically, we run 600 iterations for `stage_1`, while in the full two-stage algorithms we restrict Stage 1 to 200 iterations. As shown in Figure 7, all methods exhibit approximately linear scaling with respect to the dimension $d$, consistent with the theoretical complexity of the underlying procedures.

Our proposed algorithms are slightly slower than some baselines in these experiments. This is mainly because we do not aggressively tune hyperparameters for runtime optimization, and instead use conservative settings to ensure stability across different distributions. Overall, the results confirm that the proposed methods achieve near-linear time complexity while maintaining robustness and accuracy.

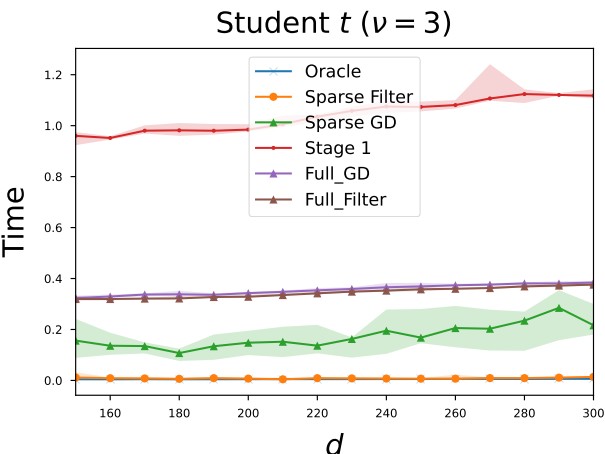

Figure 7: Running time of the proposed methods and baseline algorithms as a function of dimension $d$.

