# OpenReview forum: "Sparse Mean Estimation in Adversarial Settings via Incremental Learning"
_TMLR — Accepted by TMLR_

### Review · Reviewer_A7ys · 2026-01-11

**Summary Of Contributions:**

This paper proposes a robust sparse mean estimation under adversarial contamination and heavy-tailed distributions. Their approach follows a two-stage incremental learning framework. In the first stage, they apply a sub gradient method to identify non-zero elements. In the second stage, the framework a finer-grained estimation using previously-introduced mean estimator.

**Audience:**

Yes

**Audience Explanation:**

Robust mean estimation and learning under adversarial corruption are important topics which would gather interest in the community.

**Claims And Evidence:**

Yes

**Claims Explanation:**

The performance of the proposed framework was supported by theoretical convergence guarantees, as well as its performance on three different distributions. The authors also included performances of partial framework (just the first step) and additional ablation results in the appendix. The evaluation is complete and well-supported.

**Requested Changes:**

- Can the authors include additional details or comments on figures in the paper? Some figures, such as Figure 4 in Appendix B could benefit from additional explanation and clarification. For example, the paper attributed the weaker performance of full_filter to the suboptimal performance of sparse_GD and sparse_filter, but their performances seemed okay? It would be nice to add some additional clarification on Figure 4 and how it aligns with Figure 2.

---

> ### Author Response · Authors · 2026-02-04
> **Response to Reviewer A7ys**
>
> We sincerely thank the reviewer for the positive and thoughtful assessment of our work. We appreciate the recognition of the novelty and effectiveness of the proposed two-stage framework, as well as the careful evaluation of both the theoretical guarantees and empirical results. We are also grateful for the reviewer’s acknowledgment of the relevance of this problem to the TMLR community. Below, we respond to the specific comments and suggestions in detail.
>
> > Can the authors include additional details or comments on figures in the paper? Some figures, such as Figure 4 in Appendix B could benefit from additional explanation and clarification. For example, the paper attributed the weaker performance of full_filter to the suboptimal performance of sparse_GD and sparse_filter, but their performances seemed okay? It would be nice to add some additional clarification on Figure 4 and how it aligns with Figure 2.
>
> Thank you for the suggestion. We agree that Figure 4 would benefit from additional clarification. Indeed, as the reviewer noted, both **sparse\_GD** and **sparse\_filter** perform reasonably well in Figure 4. However, as shown in Figure 3, their performance is sensitive to the choice of the sparsity parameter $k'$, and in Figure 5, we further observe that their performance degrades noticeably under heavier-tailed distributions.
>
> In contrast, our method remains stable across these settings without requiring accurate prior knowledge of $k$. We have added a clarifying discussion in the revised manuscript to better explain these observations and to explicitly connect the behaviors observed in Figures 3–5 with the trends shown in Figure 4.

---

### Review · Reviewer_xUQq · 2026-01-21

**Summary Of Contributions:**

In this paper, the authors study the problem of estimating a k-sparse mean from contaminated samples drawn from an unknown heavy-tailed distribution. To address this challenge, they propose an efficient estimator that achieves robust and accurate mean estimation under both sparsity and contamination.

**Audience:**

Yes

**Audience Explanation:**

This work addresses finding an estimator for a k-spare mean of an unknown distribution, which has applications in the machine learning area. Therefore, it would be interesting for the TMLR's audience.

**Claims And Evidence:**

Yes

**Claims Explanation:**

Overall, the claims and contributions appear convincing and are clearly articulated. However, I found the discussion regarding the assumption of an unknown distribution somewhat unclear. In particular, based on my understanding of the experimental section, all the distributions considered there seem to be known. Additionally, Step 2 of the algorithm is described as optional. However, the experimental results indicate that the performance of the first stage alone is inferior to that of other baselines. This raises some confusion as to whether Step 2 should truly be considered optional or instead a necessary component of the method. On the other hand, to demonstrate that an estimator performs well, it is typically necessary to establish unbiasedness and vanishing variance. In Theorem 1, however, I found it unclear how these two properties are proven.

**Requested Changes:**

1. The paper would benefit from a preliminary section that introduces and clarifies the notation used throughout. For instance, the Hadamard product appearing in Equation (2) is not defined, and the parameter k representing the level of sparsity is introduced only after it has already been used earlier in the text.
2. Some of the plots are confusing. For example, Figure 1 lacks a proper legend/explanation, making it unclear which color represents what.
3. The Oracle estimator used in the experiments in the main paper is not well introduced; it is only described later in the appendix.
4. Figure 7 appears to show only the running time of your proposed methods. It would be more informative to include a broader comparison with other relevant baselines

---

> ### Author Response · Authors · 2026-02-04
> **Response to Reviewer xUQq (Part I)**
>
> We thank the reviewer for the careful reading of our manuscript and for the constructive and encouraging feedback. We address each point below.
>
> > However, I found the discussion regarding the assumption of an unknown distribution somewhat unclear. In particular, based on my understanding of the experimental section, all the distributions considered there seem to be known.
>
> We clarify that “unknown distribution” means the data-generating distribution is **unknown to the algorithm**, rather than arbitrary beyond the stated assumptions. Our theory only relies on mild moment conditions (unknown mean, bounded covariance, bounded coordinate-wise third moment) together with adversarial contamination under the strong contamination model; the estimator itself does not use any knowledge of the underlying distribution.
>
> In the experiments, we evaluate the method on specific heavy-tailed distributions (Fisk, Pareto, Student-t) purely for benchmarking and illustration. These choices are used to demonstrate robustness across different tail behaviors and do not contradict the assumption that the distribution is unknown to the algorithm.
>
> > Additionally, Step 2 of the algorithm is described as optional. However, the experimental results indicate that the performance of the first stage alone is inferior to that of other baselines. This raises some confusion as to whether Step 2 should truly be considered optional or instead a necessary component of the method.
>
> This is an important point, and we thank the reviewer for raising it.
>
> - **Stage 1** is primarily designed to provide a *coarse-grained estimate* and, in particular, to enable **support identification via incremental learning**. Under the general moment conditions and the strong contamination model, Stage 1 guarantees an $\ell_2$-estimation error of order $O(\sigma\sqrt{k\varepsilon})$ for a suitable range of iterations (Theorem 1, first bullet).
>
> - **Stage 2** refines the estimate of the recovered support to achieve the **optimal statistical rate** $O(\sigma\sqrt{\varepsilon})$. However, the *success of Stage 2 relies on correctly recovering the support*, which in turn requires a **stronger signal-to-noise ratio (SNR) assumption** on the contamination rate. Concretely, the support identification guarantee in Theorem 1 requires an additional condition of the form $\varepsilon \lesssim \mu_{\min}^{\star 2}/\sigma^2$ (Theorem 1, second bullet), and Theorem 2 (end-to-end guarantee) inherits this requirement.
>
> With this in mind, Stage 2 is described as *optional* in the sense that:
>
> - Stage 1 alone already provides a meaningful estimator and a near-linear-time procedure under more general conditions (without requiring the additional SNR condition needed for certified support recovery);
> - Stage 2 is only invoked when one aims to achieve the optimal rate, which requires the stronger contamination/SNR condition to ensure correct support recovery and thus enable dimension reduction.
>
> The experiments show that Stage 1 alone can be inferior in the final $\ell_2$ error compared to methods that effectively perform refined estimation, which is consistent with the theory: Stage 1 is a coarse estimation/identification stage, while the optimal-rate accuracy is obtained after refinement on the support.
>
> To address the reviewer’s comment, we clarify this important distinction in the revised paper.
>
> > On the other hand, to demonstrate that an estimator performs well, it is typically necessary to establish unbiasedness and vanishing variance. In Theorem 1, however, I found it unclear how these two properties are proven.
>
> Thank you for raising this point, which highlights an important distinction in the strong contamination setting.
>
> Under the strong contamination model, **unbiased estimation is information-theoretically impossible**, since an adversary may arbitrarily replace an $\varepsilon$-fraction of the samples. As a result, no estimator can be unbiased, even asymptotically. For this reason, standard robust mean estimation results do not analyze unbiasedness or vanishing variance, but instead focus on **finite-sample minimax error bounds**.
>
> Our analysis follows this paradigm. Theorem 1 provides a high-probability finite-sample error guarantee, with an unavoidable error floor of order $\Theta(\sqrt{\varepsilon})$ due to contamination, together with a sample-dependent term. Variance enters the analysis through bounded covariance and moment assumptions, which are used to control concentration and the subgradient dynamics, rather than through asymptotic variance calculations.

---

> > ### Author Response · Authors · 2026-02-04
> > **Response to Reviewer xUQq (Part II)**
> >
> > > The paper would benefit from a preliminary section that introduces and clarifies the notation used throughout. For instance, the Hadamard product appearing in Equation (2) is not defined, and the parameter k representing the level of sparsity is introduced only after it has already been used earlier in the text.
> >
> > We thank the reviewer for this suggestion. We would like to clarify that a **Notation** subsection is already included at the end of Section 2, where we define the Hadamard product ($\odot$), projection operators, vector norms, indicator functions, and related notation.
> >
> > > The Oracle estimator used in the experiments in the main paper is not well introduced; it is only described later in the appendix.
> >
> > Thank you for pointing this out. We have now explicitly defined the oracle estimator on page 14 of the revised manuscript.
> >
> > >  Figure 7 appears to show only the running time of your proposed methods. It would be more informative to include a broader comparison with other relevant baselines.
> >
> > Thank you for the suggestion. We have updated Figure 7 to include a broader set of relevant baselines. As shown in the revised figure, all methods exhibit an approximately linear scaling in the dimension $d$. Our proposed algorithms are slightly slower in these experiments, which is mainly due to the fact that we did not aggressively tune or optimize hyperparameters for runtime performance. We have added a clarifying discussion in Appendix B.6 to explain this comparison and the observed trends.

---

> > ### Comment · Reviewer_xUQq · 2026-04-05
> > **Feedback on authors' comment**
> >
> > I have carefully read the authors' explanations in Parts I and II. I thank the authors for clarifying all the confusions and have already revised the manuscript. They addressed my concerns and questions. I recommend this paper as an accept.

---

### Review · Reviewer_8BLy · 2026-01-30

**Summary Of Contributions:**

The paper introduces a novel two-stage framework for robust sparse mean estimation, focusing on heavy-tailed distributions and adversarial corruptions. Stage 1 provides  estimation to identify the top-$k$ nonzero elements without needing prior knowledge of the sparsity level , and Stage 2 then leverages these identified variables to utilize existing robust mean estimation methods—which traditionally require the sparsity level $k$ to be known—to achieve a finer, dimension-reduced estimation.

**Strength**

**(S1.)**
Unlike conventional sparse mean estimation methods that require prior knowledge of sparsity levels $k$, this paper proposes an approach to estimate and identify $k$-sparcity means without knowing $k$ beforehand. The validity of this process, in which most of the estimated mean components remain close to zero, has been theoretically demonstrated by ** Theorem 1**.

**(S2.)**
In this paper, Stage 2 simply applies the set of variable indexes identified in Stage 1 to existing robust average estimation methods that traditionally require known $k$. By utilizing only the variables selected in Stage 1, this framework reduces the effective dimension from $d$ to $k$. This saves significant time and memory, operating in $\tilde{O}(d)$ time in contrast to previous methods requiring $exp(d)$ or $poly(d)$ time. Furthermore, applying established robust estimation theorems to this reduced support allows the error bound to be tightened from the first stage's $O(\sigma\sqrt{k\epsilon})$ to the information-theoretically optimal $O(\sigma\sqrt{\epsilon})$.

**(S3.)**
The authors conducted extensive simulations using representative heavy tail distributions such as **Fisk, Pareto, Student's **, and the results strongly support the two aforementioned strengths. Experimental results show that the sub-gradient method managed to accurately recover the real set of indexes even when up to $30%$ of the samples were compromised. Moreover, while the errors of the first-stage algorithm increase with sparsity level, the overall second-stage algorithm maintains a stable level of error with increasing levels, perfectly consistent with theoretical predictions.

**Weakness**

**(W1.)**
According to Theorem 1, the iteration count $T$ required to guarantee convergence is defined within the range $[\frac{2}{\eta}\log(1/\alpha), \frac{6}{\eta}\log(1/\alpha)]$. This implies that setting the stepsize $\eta$ and initialization scale $\alpha$ to sufficiently small values to enhance stability and precision significantly increases the number of iterations needed for theoretical guarantees. + Additional Explanation: Although the paper reports satisfactory performance using $T=600$ in its core simulations , the learning dynamics shown in Figure 1(a) suggest that signal coefficients may actually require 1,000 iterations or more to reach a stable steady-state. This indicates that in practical high-dimensional settings where high accuracy is paramount, the computational burden to satisfy theoretical bounds could be considerably higher than the defaults used in the experiments.

**(W2.)**
Theory predicts an $O(\sqrt{\epsilon})$ error rate, but empirical results in Figure 6 show a nearly linear dependence on $\epsilon$. This discrepancy, attributed to the non-adversarial nature of the noise model used in testing, leaves the algorithm's performance under "truly" adversarial worst-case scenarios partially unverified.

**Audience:**

Yes

**Audience Explanation:**

The paper addresses a critical limitation in robust sparse mean estimation where existing methods require prior knowledge of the sparsity level. Most existing methods need to know the sparsity level  in advance. This paper proposes a new method that identifies the support of the mean without needing this input. They validate this approach by proving the "incremental learning" phenomenon mathematically. They demonstrate that a sub-gradient method can learn nonzero components sequentially while suppressing noise. This effectively solves the dependency on unknown sparsity parameters that limited previous studies. These findings on automatically identifying sparsity will be interesting to researchers working on high-dimensional robust statistics.

The two-stage framework presented in the paper offers practical value for algorithmic robust statistics. By utilizing a rough estimation to reduce the effective dimension, the proposed estimator operates in near-linear time and memory ($\tilde{\mathcal{O}}(d)$). This efficiency allows the method to overcome the conjectured computational-statistical tradeoffs that render existing polynomial or exponential time approaches impractical for high-dimensional settings. The paper shows that such speed is possible without sacrificing the optimal error rate. This provides useful lessons and a strong baseline for algorithm designers in the audience.

**Broader Impact Concerns:**

There are no specific ethical concerns or broader impact implications associated with this work.

**Claims And Evidence:**

Yes

**Claims Explanation:**

On the theoretical side, the authors provide clear mathematical proofs for their main arguments. **Theorem 1** proves that their first stage can correctly identify the non-zero variables without overfitting to noise. **Theorem 2** shows that the complete two-stage process achieves the optimal error rate. These derivations logically support their claim that the method works efficiently under the stated conditions.

The experiments also provide convincing evidence. The authors tested their method on difficult heavy-tailed distributions like Fisk and Pareto. Figure 1 visually confirms the "incremental learning" claim, showing clearly that signal coefficients grow faster than the noise. Additionally, Figure 2 validates that the method remains robust even when a some portion of the data is corrupted, matching their theoretical predictions.

**Requested Changes:**

**(For Acceptance)**

The statement of **Definition 1** could be made more precise to strictly align with the mathematical rigor of the subsequent theorems. Specifically, could you explicitly define the valid range for  (e.g., is it strictly bounded below 1/2)? Furthermore, since the adversary replaces  samples with arbitrary values to maximize the estimation error, it would be beneficial to explicitly state that the resulting dataset represents a "worst-case" scenario within the real number space. Clarifying these details would strengthen the logical connection between the definition of the corruption model and the worst-case error bounds derived in the main theorems.

In the **Theorem 1**, the error bound is established under the strong contamination model described in **Definition 1**. Since the dataset  is contaminated by an arbitrarily powerful adversary, the estimator  is a function of this specific, potentially worst-case dataset. To be mathematically rigorous, shouldn't the error bound be explicitly framed as a supremum over all possible -corrupted datasets generated by the adversary (e.g.,  $\sup_{S} \|\hat{\mu}(S) - \mu^*\|_2 \leq \dots$)? I suggest adding this formalism or clarifying why the current notation sufficiently captures the adversarial nature of the guarantee.

**(For Strengthening the work)**

I am curious about the rationale behind choosing the Subgradient Method (SubGM) for the nonconvex optimization problem in Stage 1. Majorization-Minimization (MM) algorithms are often preferred for robust optimization. Could you discuss why SubGM was selected over MM? Additionally, given the nonconvex nature of the objective function, are there specific convergence guarantees for SubGM in this setting that ensure it does not get trapped in poor local minima before the signal is identified?

While **Theorem 1** provides detailed theoretical bounds, providing more high-level intuition would greatly benefit the reader. Specifically, an intuitive explanation of why the imposed conditions (such as the bound on the coordinate-wise third moment) are sufficient to guarantee this "incremental learning" phenomenon would make the main contribution more accessible.

---

> ### Author Response · Authors · 2026-02-04
> **Response to Reviewer 8BLy (Part I)**
>
> We thank the reviewer for the thoughtful and positive assessment of our work. We appreciate the recognition of the novelty and effectiveness of the proposed two-stage framework, particularly its ability to identify sparse structure without prior knowledge of the sparsity level, its computational efficiency, and its strong empirical performance under heavy-tailed and corrupted settings. We have carefully addressed all comments and clarified the theoretical assumptions, algorithmic design choices, and empirical observations accordingly. We believe these revisions further strengthen the paper and improve its clarity and rigor.
>
> > **(W1.)** According to Theorem 1, the iteration count $T$ required to guarantee convergence is defined within the range $\left[\frac{2}{\eta} \log (1 / \alpha), \frac{6}{\eta} \log (1 / \alpha)\right]$. This implies that setting the stepsize $\eta$ and initialization scale $\alpha$ to sufficiently small values to enhance stability and precision significantly increases the number of iterations needed for theoretical guarantees. + Additional Explanation: Although the paper reports satisfactory performance using $T=600$ in its core simulations , the learning dynamics shown in Figure 1(a) suggest that signal coefficients may actually require 1,000 iterations or more to reach a stable steady-state. This indicates that in practical high-dimensional settings where high accuracy is paramount, the computational burden to satisfy theoretical bounds could be considerably higher than the defaults used in the experiments.
>
> Thank you for the detailed observation. The reviewer is spot on in noting that smaller values of \alpha and \eta result in a larger number of iterations. This trade-off is both inevitable and expected. Indeed, smaller values of $\alpha$ and $\eta$ lead to a tighter final error, which naturally requires running the algorithm for more iterations. This behavior is not specific to our method; rather, it is a common phenomenon in iterative algorithms, where achieving higher accuracy typically comes at the cost of increased computational effort.
>
> Furthermore, the apparent discrepancy in our experiments arises from the fact that different hyperparameter regimes are used in Figure 1 and in the simulation section.
>
> In Figure 1, we intentionally choose a smaller initialization scale $\alpha = 10^{-10}$ in order to illustrate the theoretical convergence behavior predicted by Theorem 1. Under this setting, the required iteration count scales as $T = \Theta\big(\frac{1}{\eta}\log(1/\alpha)\big)$, which naturally leads to a larger number of iterations before full stabilization.
>
> In contrast, in the simulation section, we use a more practical configuration $\alpha = 10^{-5}$, which significantly reduces the required iteration count while still achieving accurate recovery.
>
> We clarified this important distinction in the revised paper.
>
> > **(W2.)** Theory predicts an $O(\sqrt{\epsilon})$ error rate, but empirical results in Figure 6 show a nearly linear dependence on $\epsilon$. This discrepancy, attributed to the non-adversarial nature of the noise model used in testing, leaves the algorithm's performance under "truly" adversarial worst-case scenarios partially unverified.
>
> Thank you for your astute observation. The near-linear dependence on $\epsilon$ in Figure 6 arises because, as the reviewer correctly pointed out, the corruption used in the experiments is not fully adversarial. Our approach is inspired by the common practice in prior work (for example, Cheng et. al.), which typically adopts a structured (non-adaptive) corruption model for empirical evaluation. This corruption approach is milder than the worst-case adversarial setting assumed in the theory. As a result, the observed error scaling is better than the conservative $O(\sqrt{\epsilon})$ worst-case bound in Theorem 1.
>
> A promising direction for future work is to examine the performance of the proposed methods under truly adversarial corruptions. We believe that constructing such adversarial examples is far from trivial (see, e.g., Shafieezadeh-Abadeh (2023)) and beyond the scope of this paper, but it is an interesting problem in its own right.
>
> This point is clarified in the revised paper.

---

> > ### Author Response · Authors · 2026-02-04
> > **Response to Reviewer 8BLy (Part II)**
> >
> > > The statement of **Definition 1** could be made more precise to strictly align with the mathematical rigor of the subsequent theorems. Specifically, could you explicitly define the valid range for (e.g., is it strictly bounded below 1/2)? Furthermore, since the adversary replaces samples with arbitrary values to maximize the estimation error, it would be beneficial to explicitly state that the resulting dataset represents a "worst-case" scenario within the real number space. Clarifying these details would strengthen the logical connection between the definition of the corruption model and the worst-case error bounds derived in the main theorems.
> >
> > Thank you for this helpful suggestion. We agree that clarifying the admissible range of the corruption parameter improves precision and readability.
> >
> > In Definition 1, we intentionally keep the corruption level general and denote it by an abstract constant $\epsilon_0$, since the admissible upper bound on the corruption rate depends on the specific guarantees being established. To avoid over-restricting the model at the definition stage, we therefore do not impose a fixed numerical bound (e.g., $\epsilon < 1/2$) there.
> >
> > That said, in the main results—including **Theorem 1**, **Proposition 2**, and **Theorem 2**—we explicitly specify the required upper bounds on the corruption parameter. These bounds are stated clearly (highlighted in blue in the revised manuscript) and are sufficient for the corresponding guarantees to hold. In all cases, the adversarial corruption is treated in a worst-case manner, where the adversary may replace samples with arbitrary values in $\mathbb{R}^d$.
> >
> > > In the **Theorem 1**, the error bound is established under the strong contamination model described in **Definition 1**. Since the dataset is contaminated by an arbitrarily powerful adversary, the estimator is a function of this specific, potentially worst-case dataset. To be mathematically rigorous, shouldn't the error bound be explicitly framed as a supremum over all possible -corrupted datasets generated by the adversary (e.g., $\sup _S\left|\hat{\mu}(S)-\mu^*\right|_2 \leq \ldots$)? I suggest adding this formalism or clarifying why the current notation sufficiently captures the adversarial nature of the guarantee.
> >
> > Thank you for this helpful comment. We agree that, under the strong contamination model, the estimator should be interpreted as operating on a worst-case corrupted dataset.
> >
> > In our setting, this worst-case nature is already implicit in Definition 1: the adversary may arbitrarily replace an $\epsilon$-fraction of samples, and the probability statements in Theorem 1 are taken only over the randomness of the uncontaminated data. The estimator therefore depends on a potentially adversarial dataset, and the guarantees hold uniformly over all admissible corruptions.
> >
> > To make this explicit and improve clarity, we have revised the statement of Theorem 1 to emphasize that the bound holds for *any* dataset generated under the strong contamination model, i.e., uniformly over all $\epsilon$-corrupted samples. This makes the worst-case nature of the guarantee explicit while keeping the statement concise.
> >
> > > I am curious about the rationale behind choosing the Subgradient Method (SubGM) for the nonconvex optimization problem in Stage 1. Majorization-Minimization (MM) algorithms are often preferred for robust optimization. Could you discuss why SubGM was selected over MM? Additionally, given the nonconvex nature of the objective function, are there specific convergence guarantees for SubGM in this setting that ensure it does not get trapped in poor local minima before the signal is identified?
> >
> > Thank you for the question. While MM-type methods are indeed effective for many robust optimization problems, our use of SubGM is intentional and serves a different purpose.
> >
> > In our setting, Stage 1 is **not aimed at minimizing the nonconvex objective to optimality**. In fact, in the regime $n<d$, the globally optimal solution is likely to overfit to noise. Instead, it is designed as a *signal identification* step. MM methods tend to aggressively fit the data and may quickly overfit to adversarial corruptions, which is undesirable in the strong contamination regime.
> >
> > By contrast, SubGM is used in an **early-stopping regime**, where its incremental updates allow the algorithm to amplify the true sparse signal while avoiding premature fitting of corrupted samples. Our analysis shows that within a controlled number of iterations, SubGM reaches a region where the signal is correctly identified and the estimation error is bounded at the desired rate.
> >
> > Therefore, SubGM is not used as a global optimizer here, but rather as a mechanism for separating signal from noise prior to refinement.

---

> > > ### Author Response · Authors · 2026-02-04
> > > **Response to Reviewer 8BLy (Part III)**
> > >
> > > > While **Theorem 1** provides detailed theoretical bounds, providing more high-level intuition would greatly benefit the reader. Specifically, an intuitive explanation of why the imposed conditions (such as the bound on the coordinate-wise third moment) are sufficient to guarantee this "incremental learning" phenomenon would make the main contribution more accessible.
> > >
> > > Thank you for the suggestion. We agree that additional intuition is helpful.
> > >
> > > The requirement on the coordinate-wise third moment is mainly **technical**, arising from the use of a Berry–Esseen–type bound to control empirical deviations under contamination. It ensures sufficient concentration so that, in the early iterations, the algorithm consistently amplifies the true signal while noise and adversarial effects remain controlled.
> > >
> > > Importantly, this condition is **not fundamental to the incremental learning phenomenon itself**. We believe the third-moment assumption could be relaxed or removed using more refined analytical tools, and we have clarified this point in the revised manuscript.

---

### Decision · Action_Editor_4c7R · 2026-04-27

**Recommendation:** Accept as is

**Audience:**

Yes

**Audience Explanation:**

The paper could be of interest to researchers working on robust statistics / robust machine learning, especially high-dimensional robust estimation as well as researchers focus on Optimization for statistical learning.

**Claims And Evidence:**

Yes

**Claims Explanation:**

This paper studies the problem of sparse mean estimation under adversarial corruptions. In particular, it proposes a simple, scalable estimator that does not require prior knowledge of the sparsity level k and is applicable in high-dimensional settings.
The paper theoretically establishes a near-optimal error bound and demonstrates near-linear time and memory complexity, effectively validating the methodology’s soundness.

In addition, the authors adequately answered the reviewers' comments.